# *Plasmodium* ARK2 and EB1 drive unconventional spindle dynamics, during chromosome segregation in sexual transmission stages

Mohammad Zeeshan[1], Edward Rea[1], Steven Abel[2], Kruno Vukušić [3], Robert Markus[1], Declan Brady[1], Antonius Eze [1,9], Ravish Rashpa [4], Aurelia C. Balestra [4], Andrew R. Bottrill [5], Mathieu Brochet [4], David S. Guttery[6], Iva M. Tolić [3], Anthony A. Holder [7], Karine G. Le Roch [2], Eelco C. Tromer [8] & Rita Tewari [1] ✉

The Aurora family of kinases orchestrates chromosome segregation and cytokinesis during cell division, with precise spatiotemporal regulation of its catalytic activities by distinct protein scaffolds. *Plasmodium* spp., the causative agents of malaria, are unicellular eukaryotes with three unique and highly divergent aurora-related kinases (ARK1-3) that are essential for asexual cellular proliferation but lack most canonical scaffolds/activators. Here we investigate the role of ARK2 during sexual proliferation of the rodent malaria *Plasmodium berghei*, using a combination of super-resolution microscopy, mass spectrometry, and live-cell fluorescence imaging. We find that ARK2 is primarily located at spindle microtubules in the vicinity of kinetochores during both mitosis and meiosis. Interactomic and co-localisation studies reveal several putative ARK2-associated interactors including the microtubule-interacting protein EB1, together with MISFIT and Myosin-K, but no conserved eukaryotic scaffold proteins. Gene function studies indicate that ARK2 and EB1 are complementary in driving endomitotic division and thereby parasite transmission through the mosquito. This discovery underlines the flexibility of molecular networks to rewire and drive unconventional mechanisms of chromosome segregation in the malaria parasite.

Cell division proceeds through either mitosis or meiosis, after DNA replication to enable eukaryotes to propagate, proliferate and evolve in diverse ecological niches[1]. Aurora kinases (AKs) are a conserved family of protein kinases, with critical roles in four aspects of cell division at different locations: (I) driving mitotic/meiotic spindle assembly and disassembly, (II) regulating spindle pole structure and dynamics, (III) promoting accurate chromosome segregation, and (IV) orchestrating cellular fission at cytokinesis[2,3] (Fig. 1A). While the Last

Eukaryotic Common Ancestor (LECA) executed all these regulatory functions with a single AK, widespread gene duplication produced variable numbers of paralogues in diverse eukaryotic subgroups[4]. Many eukaryotic lineages have retained the singular ancestral AK, including baker's yeast, *Saccharomyces cerevisiae* (Ipl1)[5], the slime mould *Dictyostelium discoideum* (aurK)[6] and the intestinal parasite *Giardia intestinalis*[7,8]. *Caenorhabditis* spp. (air-1 and 2) and *Drosophila* spp. (aurA and B) have two AKs[9]. Some lineages have three AKs: for

example, mammals (Aurora A to C)[9], flowering plants (Aurora 1 to 3)[10], kinetoplastid parasites (AUK 1 to 3)[11] and apicomplexan parasites (ARK1 to 3)[12,13] (Fig. 1B and Supplementary Data 1).

How, when and which functions are executed by each of the Aurora paralogues (either single or multiple) varies extensively between these eukaryotic lineages (summarised in Supplementary Fig. 1 and Supplementary Note 1). One paralogue (in humans, Aurora A) is called the "polar aurora", due to its association with centrosomal subunits including Cep192 and the microtubule (MT)-assembly factor TPX2[3,9], which govern the spindle (pole)-specific location. A second paralogue (in humans, Aurora B) has been designated the "equatorial aurora" as it localises to the midplane of a dividing cell to regulate chromosome bi-orientation on the metaphase spindle to mediate cytokinesis at the last stage of cell division[2–4]. This second paralogue associates with the chromosomal passenger complex (CPC), a heterotrimeric scaffold (comprised of INCENP, Survivin and Borealin) that provides local AK activity at inner centromeres and kinetochores until metaphase, after which it translocates to MTs of the central spindle, to orchestrate cytokinesis[14,15]. A third paralogue provides an evolutionary platform for novelty. In humans, Aurora C is a meiosis-specific Aurora B variant[16]. In plants, kinetoplastids, and apicomplexans, the three AKs are less well studied but are likely contributing to divergent aspects of cell division in these lineages.

*Plasmodium* spp., the causative agent of malaria, belong to the phylum Apicomplexa, a group of intracellular, unicellular parasites with unusual aspects of division and multiplication. Previous phylogenetic analyses of Apicomplexa had identified three genes for Aurora Related Kinases (ARKs) 1, 2 and 3 in *Plasmodium* spp.[17] and the coccidian *Toxoplasma gondii* (Tg)[12,13]. In *Cryptosporidium* spp., only one Aurora orthologue has been identified, suggesting an expansion of the ARK family in the common ancestor of *Plasmodium* and *Toxoplasma*[13]. Broad functional characterisation of the three *Toxoplasma* ARKs identified TgARK1 as associated with the CPC component INCENP1, while TgARK2 located at centromeres[12,13]. TgARK2 and TgARK3 have been shown to localise to the spindle and spindle pole, and cleavage furrow during cytokinesis, respectively. However, molecular scaffolds and/or activators of TgARK2-3 have not yet been characterised[12,13]. Functional studies with both the human parasite *Plasmodium falciparum* and rodent parasite *Plasmodium berghei* have suggested that all three *Plasmodium* ARKs are likely essential for proliferation in asexual blood-stage schizogony[18–20]. Further characterisation of PfARK1[17] and PfARK3[13] was limited to asexual blood stages of *P. falciparum*, with PfARK1 shown to be potentially associated with spindle poles[17] and PfARK3 located next to the DNA at discrete foci[13]. Previous gene expression studies suggested that ARK2 is involved in proliferation and cell division in both *P. falciparum* and *P. berghei*[21,22], but nothing is known about the location or involvement of any *Plasmodium* scaffold/activator.

Within the mosquito host, the mitotic process differs substantially from that in asexual blood-stage schizogony, in which closed mitosis

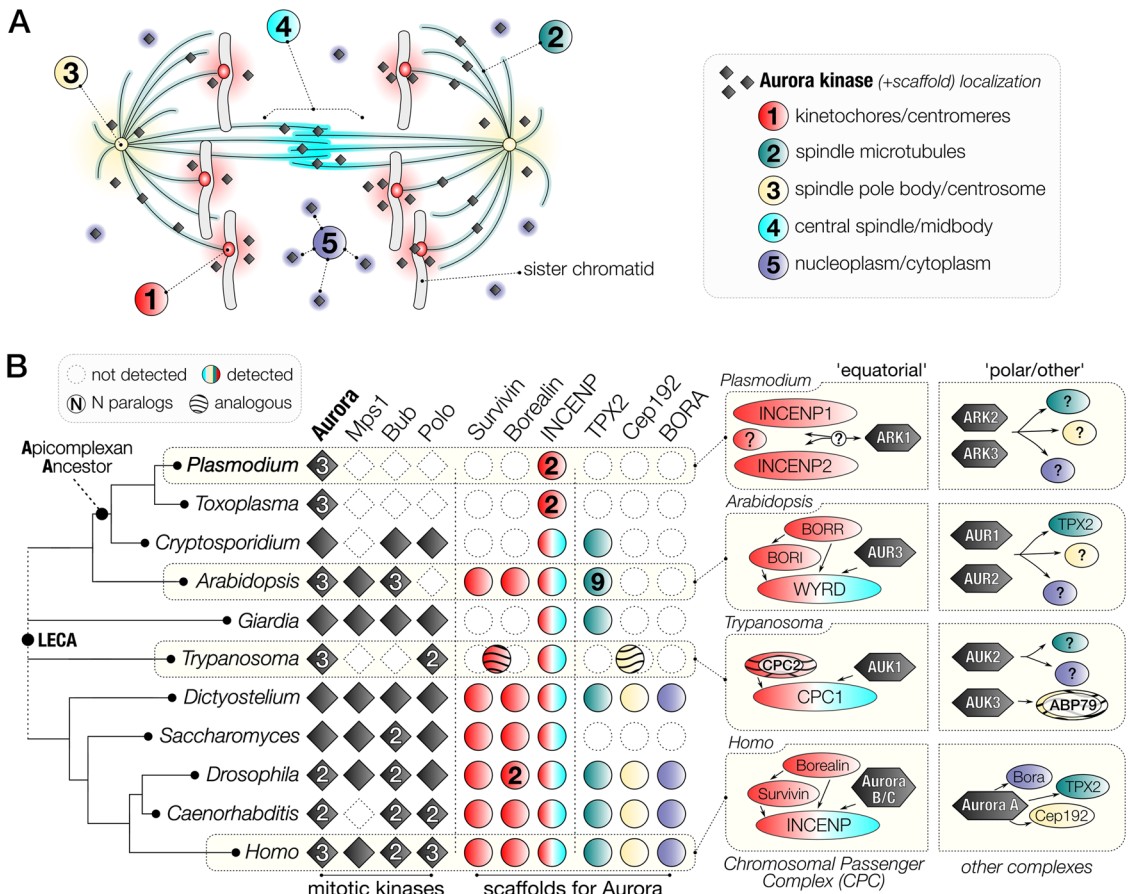

**Fig. 1 | Comparative genomics of the Aurora kinase family and known interactors indicate extensive divergence in the molecular regulation of *Plasmodium* spp. cell division. A** The five locations of Aurora kinase (AK) activity during chromosome segregation in canonical late anaphase mitotic cell progression. The same colour for each subcellular location is used in (**B**). **B** Presence-absence matrix of eukaryote mitotic kinases focused on Apicomplexa and AKs, including the scaffolds and activators of this essential kinase family. Right: known or suggested mitotic location of Aurora paralogs and their protein complexes in subgroups exemplified by model systems. A general pattern of sub-functionalisation after gene duplication within the AK family can be discerned: one paralogue interacts with the CPC: Survivin-Borealin-INCENP complex ('equatorial'), while other paralogues can be found in the cytoplasm, at the spindle pole and/or on spindle MTs ('polar/other') (see Supplementary Fig. S1). LECA Last Eukaryotic Common Ancestor.

is associated with asynchronous nuclear division that precedes cytokinesis[23–25]. During male gametogony in the mosquito gut, rapid mitosis is characterised by threefold genome replication from 1N to 8N[24,26,27]. Concomitant spindle formation and chromosome segregation occur within 8 min without nuclear division, followed by karyokinesis and cytokinesis resulting in eight haploid male gametes. Thus, mitosis during asexual stages and male gametogony differs not only in timing but also in the rounds of DNA replication and chromosome segregation[28]. The male gamete is the only flagellated form in *Plasmodium* and it finds and fertilises a female gamete, resulting in zygote formation. Meiosis commences within 24 h of fertilisation during zygote differentiation, with an initial genome duplication from 2N to 4N[24]. Further divisions occur in the subsequent oocyst, through endomitotic cycles resulting in thousands of haploid sporozoites[27,29]. Our previous studies identified an unconventional toolkit of cell cycle proteins, with mitotic protein kinases and phosphatases regulating these processes[20,30,31]. In addition to the three divergent ARKs, genes encoding seven CDK-related kinases, and four divergent Nima-like kinases have been identified in the *Plasmodium* genome. However, *Plasmodium* spp. seem to have lost many common cell division kinases such as Bub1, Mps1 and Polo, making their complement of cell division kinases quite different from that of many model eukaryotes[20,27,30] (Fig. 1B). In addition, the presence and role of scaffold proteins for *Plasmodium* ARKs is poorly understood.

Here we investigate the role and putative scaffold of ARK2, focusing on the sexual stages of *P. berghei*. Using fluorescent real-time live-cell imaging, proteomics, and functional genetic studies at distinct proliferative stages within the mosquito host, we reveal that ARK2 is located at the spindle where it is associated with spindle and kinetochore proteins that includes EB1, MISFIT, and MyoK. We find that both ARK2 and EB1 are critical components for spindle dynamics and the cycles of chromosome segregation, and therefore are crucial factors for parasite transmission.

## Results

### Spatiotemporal studies of ARK2-GFP during male gametogony demonstrate its association with rapid spindle dynamics

The spatiotemporal dynamics of ARK2 expression in real time was analysed during male gametogony using ARK2-GFP (summarised in Supplementary Fig. 2 and Supplementary Note 2). Prior to gametocyte activation, ARK2-GFP was detected as a diffuse signal within the nucleus although some gametocytes had a single concentrated focus (Fig. 2A). One minute after activation, the protein was located at a single point in the nucleus, before extending into an elongated bridge-like structure, which collapsed into two separate points within the next 1–2 min (Fig. 2B and Supplementary Movie 1). Each separate point then extended into a bridge before collapsing to form four separate foci (Supplementary Fig. 2G and Supplementary Movie 2). A repeat of this cycle resulted in eight foci, all within 8 min (Supplementary Fig. 2H and Supplementary Movie 3). Once mature male gamete formation (exflagellation) began, these foci faded, leaving a diffuse nuclear signal (Fig. 2A). These cycles of extension and collapse to individual foci were often asynchronous with respect to similar events within the same nucleus−suggesting that mitosis in male gametogony is an asynchronous form of cell division (Fig. 2A).

To examine further the location of ARK2 we investigated by indirect immunofluorescence assay (IFA) its co-localisation with MTs that had been labelled with an α-tubulin antibody, using fixed gametocytes at different times after activation. Alpha-tubulin antibody detected both nuclear mitotic spindles and developing cytoplasmic axonemes, but ARK2 colocalized only with the mitotic spindles at all stages of gametogony (Supplementary Fig. 3B). This result provides evidence that ARK2 is involved in mitosis within the male gametocyte. To improve the resolution of detection, we used deconvolution microscopy and these images suggest that ARK2 is

located on mitotic spindles during male gametogony (Supplementary Fig. 3C).

### ARK2 is associated with kinetochore (NDC80) and not axonemal (kinesin-8B) dynamics during male gametogony

To investigate further the association of ARK2 and the mitotic spindle during male gametogony, we compared ARK2 location with that of the kinetochore marker NDC80 and cytoplasmic axonemal protein kinesin-8B. We generated parasite lines expressing ARK2-GFP (green) and ARK2-mCherry (red) (Supplementary Fig. 2A) and genetically crossed them with different marker lines. Parasite lines expressing ARK2-mCherry and NDC80-GFP[29] were crossed, and the progeny were analysed by live-cell imaging to establish the spatiotemporal relationship of these two tagged proteins. Both ARK2-mCherry and NDC80-GFP were located next to the stained DNA, and with a partial overlap, although NDC80-GFP was always closer to the DNA (Fig. 2C and Supplementary Fig. 4A). This orientation and overlap (with a Pearson's colocalization coefficient (R) of more than 0.6) of ARK2-mCherry and NDC80-GFP continued throughout male gametogony. Furthermore, the bridge length of NDC80-GFP was shorter than that of ARK2-mCherry. Time-lapse imaging showed that the dynamic redistribution of ARK2-mCherry begins prior to that of NDC80-GFP and ends slightly earlier (Fig. 2D, Supplementary Fig. 4B and Supplementary Movies 4 and 5).

Parasite lines expressing ARK2-GFP and kinesin-8B-mCherry were crossed and examined by live-cell imaging of both markers. One to two minutes after gametocyte activation, ARK2-GFP was observed close to the DNA and adjacent to, but not overlapping, the kinesin-8B-mCherry tetrad (Fig. 2E and Supplementary Fig. 4C). ARK2-GFP remained distributed on spindles, while there was duplication of kinesin-8B-mCherry-labelled tetrads (Fig. 2E and Supplementary Fig. 4C). In later stages of male gametogony, ARK2-GFP remained associated with spindles and spindle poles, while kinesin-8B-mCherry showed a distinct cytoplasmic axonemal location (Fig. 2E and Supplementary Fig. 4C). This location pattern was also observed in time-lapse imaging, with no co-localisation of ARK2-GFP and kinesin-8B-mCherry (Fig. 2F, Supplementary Fig. 4D and Supplementary Movies 6 and 7). The dynamic distribution of these two proteins demonstrates that both chromosome segregation in the nucleus, tagged with ARK2, and axoneme formation in the cytoplasm, tagged with kinesin-8B begin at a very early stage of gametogony, continuing in parallel and coordinated between these different compartments of the male cell.

To examine further the location of ARK2 with reference to the spindle, axoneme and kinetochore at high resolution; we used super-resolution confocal stimulated emission depletion (STED) microscopy, ultrastructure expansion microscopy (UExM) and 3D-structured illumination microscopy (SIM) (Fig. 2G–I). STED images of fixed gametocytes labelled with anti-GFP, and anti-α-tubulin antibodies revealed the ARK2 distribution on nuclear spindle MTs (Fig. 2G and Supplementary Fig. 5A). This visualisation was further improved by UExM on fixed gametocytes labelled with anti-HA antibodies (for ARK2) and anti-α/β-tubulin antibodies (for spindle and axonemes). UExM images clearly showed the ARK2 signal overlapping with spindle MTs and not with cytoplasmic axonemal MTs (Fig. 2H and Supplementary Fig. 5B). These observations further indicate that ARK2 distributes on spindle MTs. Next, we used 3D-SIM on fixed gametocytes expressing both ARK2-mCherry and NDC80-GFP, which clearly showed the ARK2 bridge associated with punctate NDC80-labelled kinetochores (Fig. 2I and Supplementary Fig. 5C).

### Tracing ARK2-GFP location during the zygote to ookinete transition indicates a role at the meiotic spindle

To characterise the location of ARK2 in meiotic (i.e., zygote/ookinete) stages, ARK2-GFP dynamics were observed in developing ookinetes over a 24 h period. At various points of ookinete development,

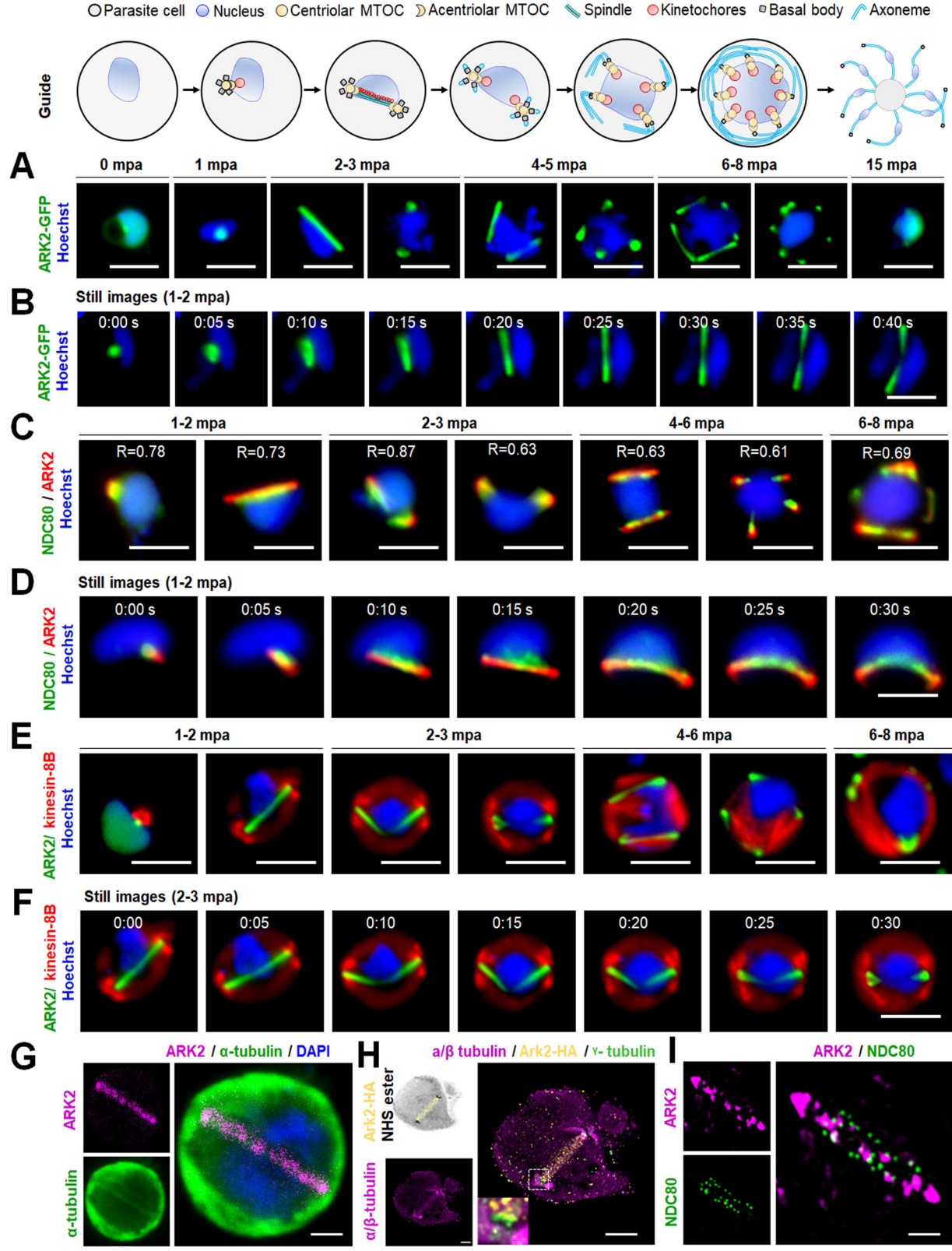

ARK2-GFP was detected as focal points like those observed during male gametogony, as well as structures radiating into the nuclear equator (Fig. 3A). In zygotes (2 h after gametocyte activation and fertilisation), ARK2-GFP was detected at one or two foci. These foci migrated away from each other over the next 8–10 h through development into stage IV ookinetes, to opposite sides of the nucleus (Fig. 3A). During this time, the ARK2-GFP signal appeared to radiate into the centre of the nucleus, typical of a classic metaphase spindle arrangement (Fig. 3A). These two foci then divided again to form four foci, before the signal faded into a diffuse distribution within nuclei of mature ookinetes (Fig. 3A). The location of ARK2 relative to that of the kinetochore marker, NDC80, was examined during ookinete development in parasite lines expressing ARK2-mCherry and NDC80-GFP. ARK2-mCherry was located on spindles radiating from the poles and

**Fig. 2 | Real-time live-cell imaging of PbARK2 reveals spindle association and kinetochore dynamics during male gametogony.** The upper schematic shows the major stages of male gametogony with subcellular structures identified. **A** Imaging of ARK2-GFP (green) during male gametogony reveals an initial location at the putative MT organising centre (MTOC) just after activation (1 min post activation; mpa), and at the spindles and spindle poles in later stages. The protein accumulates diffusely in the residual nuclear body after gamete formation and is not present in the flagellate gametes (15 mpa). More than 50 images were analysed in more than five different experiments. Scale bar = 5 μm. **B** Still images (5 s intervals) showing development of an ARK2-GFP bridge from one focal point followed by further division into two halves within 1 to 2 mpa. More than ten timelapses were analysed in more than five different experiments. Scale bar = 5 μm. **C** The location of ARK2-mCherry (red) relative to kinetochore marker, NDC80-GFP (green). More than 50 images were analysed in more than 5 different experiments. Scale bar = 5 μm. R is Pearson's coefficient for co-localisation. **D** Still images (5 s intervals) of the dynamic location of ARK2-mCherry and NDC80-GFP after between 1 and 2 min of activation. More than ten timelapses were analysed in more than five different experiments. Scale bar = 5 μm. **E** The relative location of ARK2-GFP (green) and the basal body and axoneme marker, kinesin-8B-mCherry (red). More than 50 images were analysed in more than 5 different experiments. Scale bar = 5 μm. **F** Still images (5 s intervals) of the dynamic location ARK2-GFP and kinesin-8B-mCherry after between 2 and 3 min of activation. More than 10 timelapses were analysed in more than 5 different experiments. Scale bar = 5 μm. **G** Indirect immunofluorescence followed by STED confocal microscopy showing co-localisation of ARK2 (purple) and α-tubulin (green) at spindle but not at cytoplasmic MTs at 1 mpa. More than 50 images were analysed in more than three different experiments. Scale bar = 1 μm. **H** Expansion microscopy showing co-localisation of ARK2 (yellow) and α/β tubulin (purple) staining at spindle but not at cytoplasmic MTs at 1 mpa. More than 20 images were analysed in more than three different experiments. Scale bar = 1 μm. **I** 3D-SIM image showing locations of ARK2 (purple) and NDC80 (green) at 1 mpa. More than 20 images were analysed in more than 3 different experiments. Scale bar = 1 μm. DNA (blue) is stained with Hoechst dye in (**A**–**F**) and with DAPI in (**G**).

NDC80-GFP was detected along the metaphase plate (Fig. 3B) during stages I to III. By stage IV, both ARK2 and NDC80 had accumulated at spindle poles (Fig. 3B). This clearly shows the unclustered kinetochores (NDC80) during stages I to III while ARK2 is located on spindles in the same line of kinetochores with a little overlap, but later in stage IV to V the kinetochores are clustered, again co-localising with spindle poles with a significant overlap (Spearman correlation coefficients of 0.89) (Fig. 3B). This dynamic location of ARK2 (spindle) and NDC80 (kinetochore) demonstrates the chromosome segregation during meiosis that happens during ookinete differentiation.

## Conditional knockdown of ARK2 reveals a crucial role during parasite transmission

ARK2 had previously been found to be most likely essential for *P. berghei* asexual blood-stage development[20]. To examine the role of ARK2 during sexual stages we used two approaches. First we tagged the endogenous ARK2 locus with sequence encoding an HA epitope tag and an auxin-inducible degron (AID) (Supplementary Fig. 6A) to degrade the fusion protein in the presence of auxin in a parasite line expressing the TIR1 protein[32]. Although successful genetic modification was confirmed by diagnostic PCR (Supplementary Fig. 6B), addition of auxin to gametocytes did not lead to ARK2-AID/HA degradation (Supplementary Fig. 6C) and there was no detectable phenotype in male gametogony (Supplementary Fig. 6D). Second, we used a promoter trap strategy, replacing the *ark2* promoter with that of cytoadherence-linked asexual protein (CLAG−PBANKA_1400600), which is not transcribed in gametocytes[33] (Supplementary Fig. 6E). The correct genetic integration was confirmed by PCR (Supplementary Fig. 6F), and ARK2 transcription was downregulated in $P_{clag}ark2$ gametocytes as shown by qRT-PCR (Supplementary Fig. 6G). A phenotypic analysis of these *ark2*-knockdown parasites was then performed at different stages of parasite development within the mosquito.

Despite the significant reduction of ARK2 expression in $P_{clag}ark2$ gametocytes (Supplementary Fig. 6G), neither mitosis in male gamete formation (exflagellation) nor meiosis in zygote differentiation (ookinete development) was affected (Fig. 4A, B). Ookinete morphology was also not affected (Fig. 4B). However, serious defects in oocyst formation (endomitosis) were observed, with a significant reduction (up to 70%) in the number of oocysts per mosquito midgut, detectable as early as day 7 post infection, and remaining significantly lower through to day 21 (Fig. 4C). Microscopic imaging of the midguts revealed that the few oocysts present were smaller than those of wild-type parasites expressing GFP (WT-GFP) after day 7. Sporogony had been completely blocked; some parasites contained dark granules, and some had a pycnotic appearance (Fig. 4D). $P_{clag}ark2$ oocysts were significantly smaller than wild-type from day 14 onwards, not growing beyond the size observed at day 7 (Fig. 4E). There were no sporozoites in the salivary glands of $P_{clag}ark2$ parasite-infected mosquitoes,

indicating that sporozoite development had been completely blocked even though some oocysts had formed (Fig. 4F).

One explanation for the significantly reduced number of $P_{clag}ark2$ compared to WT-GFP oocysts, was reduced ookinete motility. However, when we analysed ookinete motility on Matrigel, we saw no remarkable difference in the gliding motility of $P_{clag}ark2$ (Supplementary Movie 8) compared with WT-GFP parasites (Supplementary Movie 9 and Supplementary Fig. 7A, B).

Since ARK2 is expressed in male gametocytes and parasite development is affected after fertilisation, we investigated whether the defect is due to inheritance from the male gamete. We performed genetic crosses between $P_{clag}ark2$ parasites and other mutants deficient in production of either male (Δhap2)[6] or female (Δdozi) gametocytes[34]. Crosses between $P_{clag}ark2$ and Δdozi mutants produced some normal-sized oocysts that were able to sporulate, showing a partial rescue of the $P_{clag}ark2$ phenotype (Fig. 4G). In contrast, crosses between $P_{clag}ark2$ and Δhap2 did not rescue the $P_{clag}ark2$ phenotype. These results reveal that a functional *ark2* gene copy from a male gamete is required for subsequent oocyst development.

## Transcriptome analysis of $P_{clag}ark2$ parasites reveals altered expression of genes for proteins involved in several functions, including MT-based motor activity

To explore the effect of ARK2 knockdown on the expression of other genes in gametocytes, we performed RNA-seq transcriptomic analysis of $P_{clag}ark2$ and wild-type cells immediately prior to gametocyte activation (0 min) and after exflagellation (30 min post activation). The genome-wide read coverages for the four pairs of biological replicates (WT, 0 min; WT, 30 min; $P_{clag}ark2$, 0 min; and $P_{clag}ark2$, 30 min) exhibited Spearman correlation coefficients of 0.961, 0.939, 0.972 and 0.930, respectively, validating the reproducibility of the experiment. The downregulation of *ark2* gene expression in $P_{clag}ark2$ gametocytes was confirmed by the RNA-seq analysis: the number of reads mapped to this gene was significantly decreased (Supplementary Fig. 7C).

In addition to changed ARK2 expression, we detected 446 and 102 genes that were significantly upregulated and downregulated, respectively, at adjusted P value cut-off 0.01 and $\log_2$fold change cut-off of 1, in $P_{clag}ark2$ gametocytes activated for 30 min (Fig. 4H and Supplementary Data 2). Gene ontology (GO) enrichment analysis of the upregulated transcripts identified genes involved in MT-based processes−including MT-dependent motors−together with other functions including cell division and chromosome organisation (Supplementary Fig. 7D). These differences in transcript levels revealed by RNA-seq analysis were validated by qRT-PCR, focusing on genes for proteins involved in motor activity, other AKs, kinetochore proteins and genes for proteins implicated in ookinete and oocyst development (Fig. 4I). The modulation of these genes suggests the involvement of ARK2 in mitosis in male gametocytes, although the effect was manifest only later during sporogony.

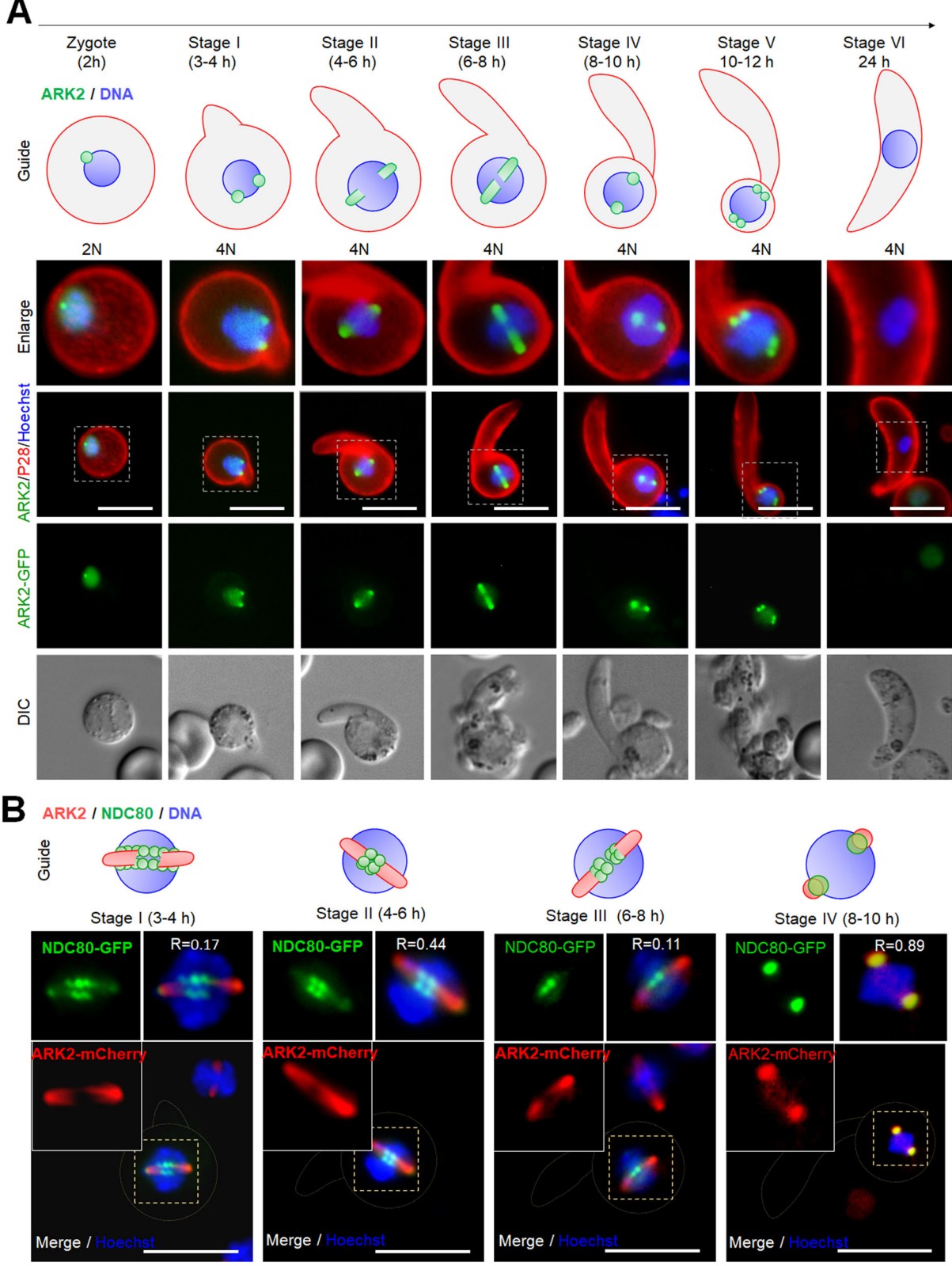

**Fig. 3 | *Pb*ARK2 is located at a putative MTOC and spindle during ookinete development.** The top schematic depicts ookinete differentiation from the zygote through six stages over a 24-h period. The genome is initially diploid (2N) and then replicated (4N) just before the nucleus migrates into the growing apical protuberance. **A** Live-cell imaging showing ARK2-GFP (green) location during ookinete development, relative to the nuclear DNA (blue, Hoechst dye), and cy3-conjugated 13.1 antibody (red), which recognises P28 protein on the surface of zygotes and ookinetes. DIC images are shown in the bottom set of panels. More than 50 images were analysed in more than three different experiments. Scale bar = 5 μm. **B** The location of ARK2–cherry (red) in relation to the kinetochore marker, Ndc80-GFP (green) and nuclear DNA (blue) at different stages of ookinete development, with a schematic guide for each stage. More than 50 images were analysed in more than three different experiments. Scale bar = 5 μm. R is Pearson's coefficient for co-localisation.

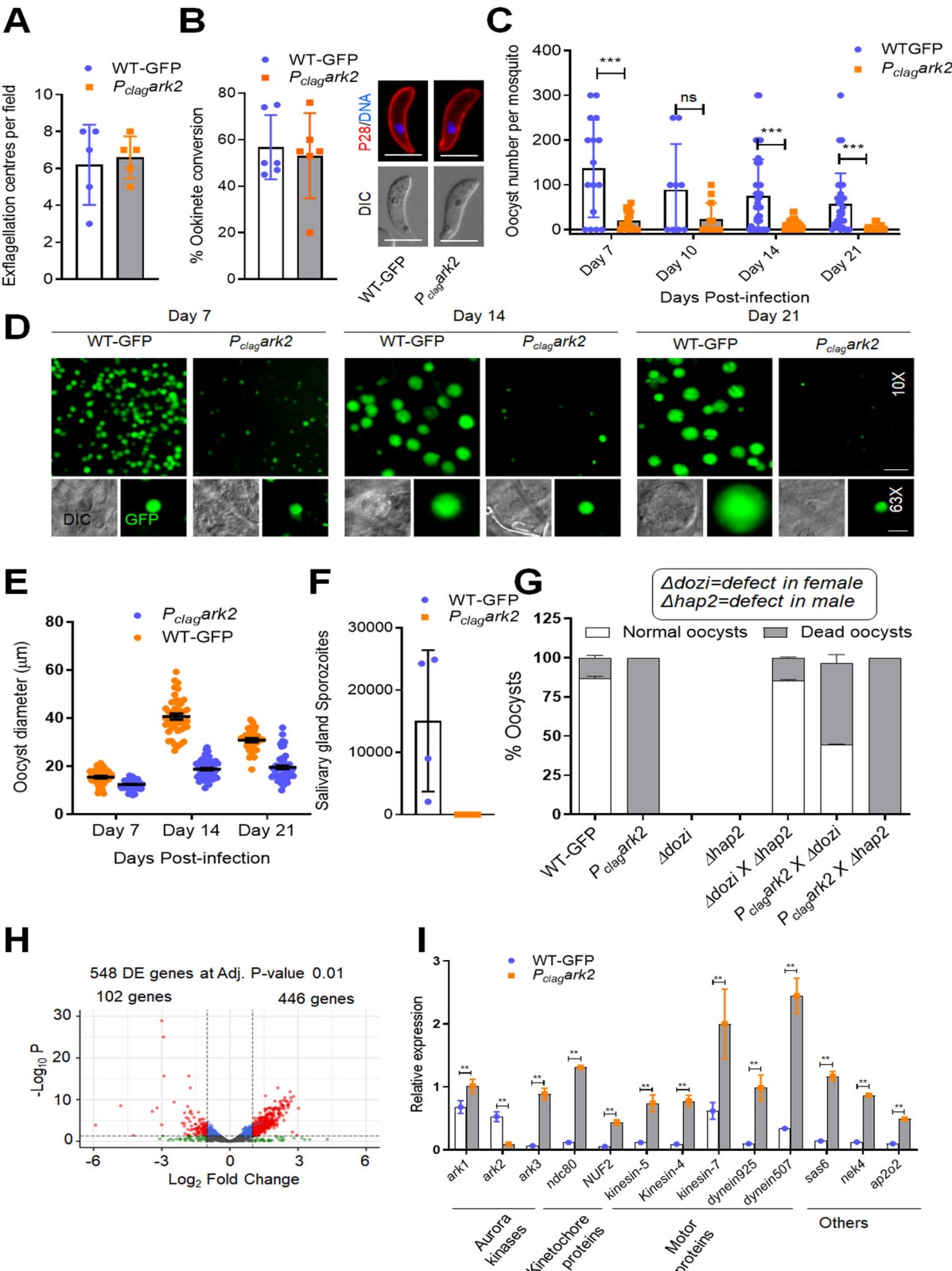

## ARK2 associates with MT-binding proteins at the spindle-kinetochore interface

Until now, no scaffold or activator proteins associated with apicomplexan ARK2 orthologues have been described. We therefore aimed to identify ARK2-interacting partners by immunoprecipitation using anti-GFP from lysates of purified gametocytes activated for 1 min when the first spindle is formed. Generally, following tryptic digestion, few peptides were identified for proteins that were uniquely present in the ARK2-GFP pulldown (compared to control WT- GFP pulldown) except for ARK2 itself, suggesting that the binding of partners to ARK2 may be transient or that the majority of ARK2 is bound to no other protein at this stage (Fig. 5 and Supplementary Data 3). Because our experiments were only performed in duplicate, conventional statistical analysis of enrichment was not possible. As an alternative, we used principal

**Fig. 4 | Conditional knockdown of PbARK2 identifies its essential role in oocyst development and sporogony. A** The number of exflagellation centres per field of $P_{clag}$-$ark2$ (black bar) compared with WT-GFP (white bar) parasites at the end of male gametogony. Shown is mean ± SD; $n = 3$ independent experiments. **B** Percentage ookinete conversion for $P_{clag}ark2$ (black bar) and WT-GFP (white bar) parasites. Ookinetes were identified by reactivity with 13.1 antibody and successful differentiation into elongated 'banana-shaped' ookinetes. Shown is mean ± SD; $n = 3$ independent experiments. Scale bar = 5 µm. **C** Total number of GFP-positive oocysts per infected mosquito in $P_{clag}ark2$ (black bar) and WT-GFP (white bar) parasites at 7, 14 and 21-day post infection (dpi). Shown is mean ± SD; $n = 3$ independent experiments (with >15 mosquitoes for each). Multiple comparison $t$ test (non-parametric) showed significant differences in oocyst number. ***$P < 0.001$; ns non-significant. ($P$ values: 0.000145−Day 7; 0.053163−Day 10; 0.000010−Day 14; 0.000193−Day 21). **D** Midguts at ×10- and ×63 magnification showing fluorescent oocysts of $P_{clag}ark2$ and WT-GFP lines at 7, 14 and 21 dpi. More than 20 images were analysed in more than three different experiments. Scale bar = 50 µm (×10) or

20 µm (×63). **E** Oocyst sizes of $P_{clag}ark2$ and WT-GFP lines at 7, 14 and 21 dpi. $n = 3$ independent experiments (with >10 mosquitoes for each and minimum 30 oocysts counted). Multiple comparisons $t$ test, with post hoc test of Holm–Sidak showed significant differences in relative expression. **$P < 0.01$, ***$P < 0.001$. The exact $P$ values and other statistics detail can be found in source data file. **F** Total sporozoite number in salivary glands of $P_{clag}ark2$ (black bar, not visible) and WT-GFP (white bar) parasites, showing mean ± SD; $n = 3$ independent experiments. **G** Rescue experiment showing male-derived allele of $P_{clag}ark2$ confers defect and is complemented by 'female' $\Delta dozi$ parasite line. Shown is mean ± SD; $n = 3$ independent experiments. **H** RNA-seq analysis showing upregulated and downregulated genes in $P_{clag}ark2$ parasites compared to WT-GFP parasites. **I** Expression level validation of relevant selected genes from the RNA-seq data using qRT-PCR. Shown is mean ± SEM; $n = 3$ independent experiments. Multiple comparisons $t$ test, with post hoc test of Holm–Sidak showed significant differences in relative expression. **$P < 0.01$, ***$P < 0.001$. The exact $P$ values and other statistics detail can be found in source data file. Source data are provided as a Source Data file.

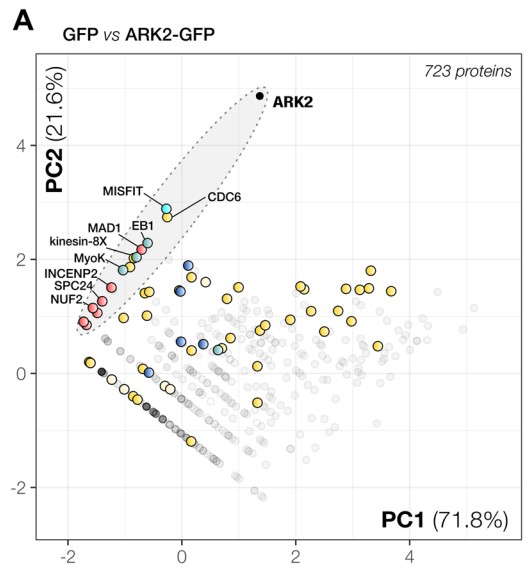

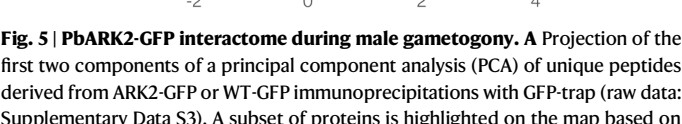

| Gene ID | Product Description | Name | AA | GFP 1 | GFP 2 | ARK2 1 | ARK2 2 |
|---|---|---|---|---|---|---|---|
| **0407400** | **serine/threonine protein kinase ARK2** | **ARK2** | **1719** | - | - | 56 | 43 |
| 1038200 | nuclear formin-like protein MISFIT | MISFIT | 1594 | - | - | 11 | 4 |
| 0405600 | microtubule-associated protein RP/EB | EB1 | 435 | - | - | 4 | 4 |
| 0908500 | myosin K | MyoK | 1629 | - | - | 4 | 1 |
| 0805900 | kinesin-8X | kinesin-8X | 1409 | - | - | 3 | 3 |
| 0612300 | conserved Plasmodium protein | MAD1 | 832 | - | - | 4 | 3 |
| 1343200 | conserved Plasmodium protein | INCENP2 | 2305 | - | - | 1 | 1 |
| 0414300 | kinetochore protein NUF2 | NUF2 | 457 | - | - | 1 | 1 |
| 1442300 | kinetochore protein SPC24 | SPC24 | 819 | - | - | 2 | 1 |
| 1102900 | cell division control protein 6 | CDC6 | 869 | - | - | 7 | 6 |
| 0312500 | origin recognition complex subunit 5 | ORC5 | 815 | - | - | 1 | 1 |
| 0501300 | DNA polymerase delta A | POLD-A | 1097 | - | - | 2 | 3 |
| 1129100 | DNA polymerase epsilon A | POLE-A | 2826 | - | - | 2 | 3 |
| 0202500 | nucleoside triphosphate hydrolase | RFC-like | 1043 | - | - | 4 | 2 |

Legend: ● spindle-associated ● kinetochore ○ MTOC ● cohesin/condensin ● DNA replication ● nuclear

**Fig. 5 | PbARK2-GFP interactome during male gametogony. A** Projection of the first two components of a principal component analysis (PCA) of unique peptides derived from ARK2-GFP or WT-GFP immunoprecipitions with GFP-trap (raw data: Supplementary Data S3). A subset of proteins is highlighted on the map based on relevant functional categories. **B** Selected proteins, their size and corresponding gene ID and representation by the number of peptides in either WT-GFP or ARK2-GFP precipitates, are presented in a table.

component analysis (PCA) to capture the co-variation of proteins over both WT-GFP and ARK2-GFP pulldown experiments (Fig. 5), a method which we have used successfully to analyse previous limited cross-linking experiments[35]. PCA revealed putative associations of ARK2 with a network of spindle MT-associated proteins including kinesin-8X (PBANKA_0805900), myosin K (PBANKA_0908500), the *Plasmodium*-specific lateral MT-binding protein EB1 (PBANKA_0405600), a variety of kinetochore proteins such as members of the NDC80 complex[29] and the recently discovered highly divergent Apicomplexan kinetochore proteins (AKiT) AKiT1-6, STU2 (PBANKA_1337500), Mad1 (PBANKA_0612300)[36], and a single peptide for the CPC subunit INCENP2 (PBANKA_1343200) (Fig. 5 and Supplementary Data 3). Interestingly, the nuclear formin-like protein MISFIT, a key regulator of ookinete to oocyst transition[37], and a putative regulator of actin filament dynamics, also appear to be part of this network of spindle-associated factors.

**Real-time imaging of parasites expressing EB1-GFP reveals its association with the spindle and kinetochore throughout male gamete formation**

Our results suggested that ARK2 is located on the spindle, possibly residing at the interface between the spindle and kinetochore, where

the unique spindle-based MT lateral-binding protein EB1 may be a central player based on protein interactomics. Therefore, we tagged EB1, encoded by the endogenous locus, with GFP (Supplementary Fig. 8A, B) and examined its spatiotemporal dynamics during male gametogony by real-time live-cell imaging. EB1-GFP showed a similar spatiotemporal distribution to that of ARK2-GFP, with distinct foci and elongated spindle 'bridges' at certain times after gametocyte activation (Fig. 6A, Supplementary Fig. 8C–E and Supplementary Movies 10–12). Chromatin immunoprecipitation with parallel sequencing (ChIP-Seq) was used to determine the DNA binding sites of EB1, and indicated its co-location with NDC80, the outer kinetochore, centromeric chromatin marker[29,38] (Fig. 6B). These results were corroborated using live-cell imaging of EB1-GFP/NDC80-mCherry dual reporter lines to show colocalization (with a Pearson's colocalization coefficient (R) of 0.95) (Fig. 6C and Supplementary Fig. 8F). EB1-GFP/ARK2-mCherry parasites showed an overlap of fluorescence signals (with a Pearson's colocalization coefficient (R) of 0.83) at 1 to 2 min post activation of gametocytes, suggesting co-location and possible interaction with spindles and the kinetochore (Fig. 6D). Finally, parasite lines expressing EB1-mCherry and the basal body marker SAS4-GFP[39] showed that the basal body (SAS4) in the cytoplasm and spindle dynamics (EB1) in the nucleus are coordinated in time and space across

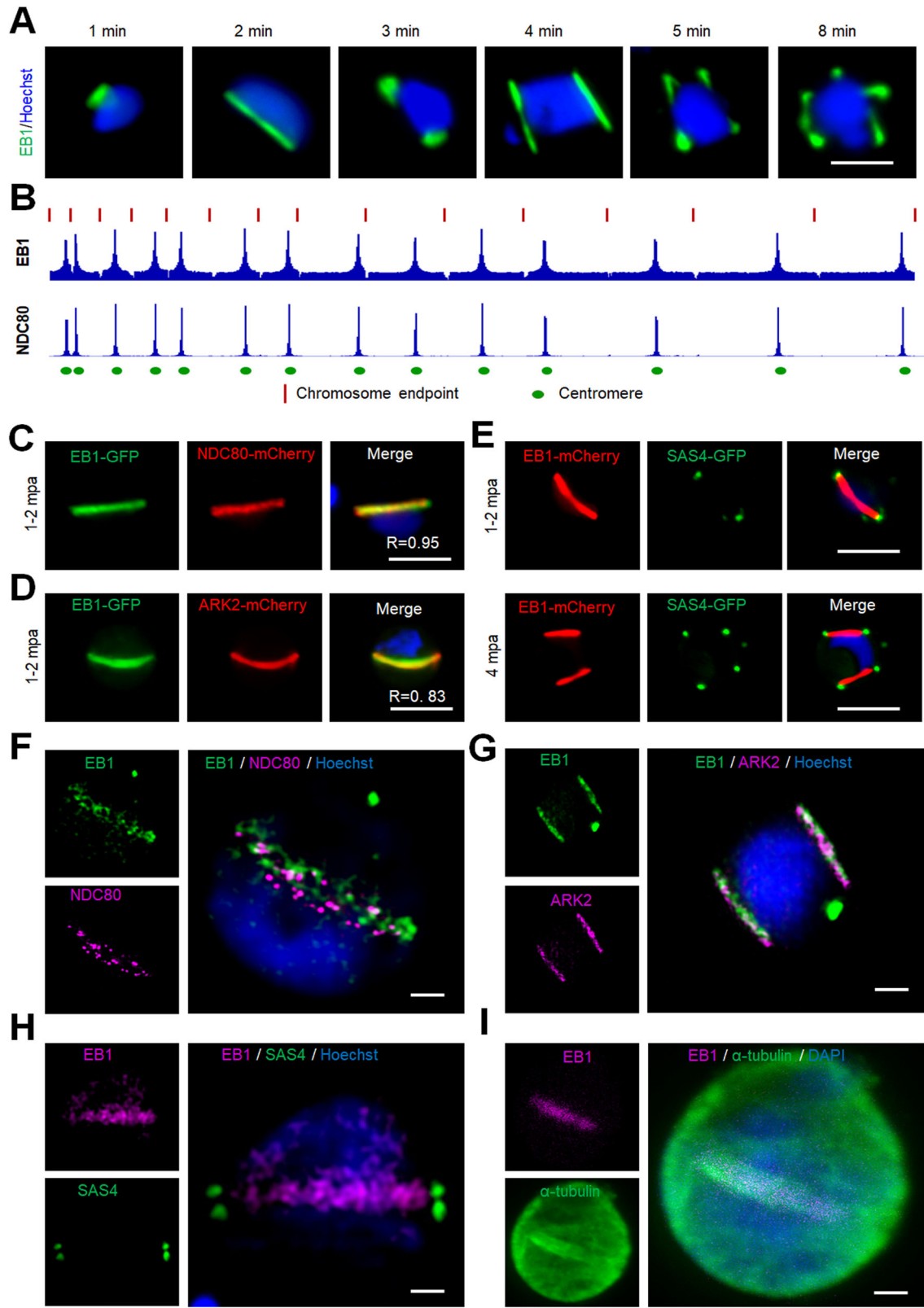

the two compartments during male gametogenesis. The basal bodies serve as the centriolar MT organising centre for axonemes in the cytoplasm (Fig. 6E).

To further resolve the location of EB1 with respect to the kinetochore and basal body at higher resolution, 3D-SIM was performed on EB1-GFP/NDC80-mCherry, EB1-GFP/ARK2-mCherry and EB1-mCherry/SAS4-GFP fixed gametocytes. The 3D-SIM images of gametocytes expressing EB1-GFP/NDC80-mCherry showed EB1 bridge(s) across the nucleus with NDC80 distributed like beads on the bridge, each bead representing at least one kinetochore (Fig. 6F and Supplementary Fig. 9A). The 3D-SIM images of gametocytes expressing EB1-GFP/ARK2-mCherry showed EB1 bridge(s) across the nucleus with ARK2, overlapping each other (Fig. 6G and Supplementary Fig. 9A). The bridged pattern of spindles for EB1 were restricted to the nucleus as shown by

**Fig. 6 | EB1; like ARK2 associates with spindle and kinetochore during male gametony. A** Live-cell imaging of EB1-GFP (green) showing its location on spindles and spindle poles. DNA is stained with Hoechst dye (blue). More than 50 images were analysed in more than three different experiments. Scale bar = 5 μm. **B** ChIP-seq analysis of EB1-GFP profiles for all 14 chromosomes showing its centromeric binding. Signals are plotted on a normalised read per million (RPM) basis. Red lines at the top indicate the ends of chromosomes; circles on the bottom indicate centromere locations. NDC80-GFP was used as a positive control and IgG was used as a negative control. **C** Live-cell imaging showing the location of EB1-GFP (green) and kinetochore marker NDC80-mCherry (red) in a gametocyte activated for 1–2 min. DNA is stained with Hoechst dye (blue). More than 50 images were analysed in more than three different experiments. Scale bar = 5 μm. R is Pearson's coefficient for co-localisation. **D** Live-cell imaging showing the location of EB1-GFP (green) and ARK2-mCherry (red) in a gametocyte activated for 1–2 min. DNA is stained with Hoechst dye (blue). More than 50 images were analysed in more than three different experiments. Scale bar = 5 μm. R is Pearson's coefficient for co-localisation. **E** Live-cell imaging showing the location of EB1-mCherry (red) and

basal body marker SAS4-GFP (green) in gametocytes activated for 1–2 min (upper panel) and 4 min (lower panel). DNA is stained with Hoechst dye (blue). More than 50 images were analysed in more than three different experiments. Scale bar = 5 μm. **F** 3D-SIM image showing location of EB1 (green) and NDC80 (purple) in a gametocyte activated for 1 min. DNA is stained with Hoechst dye (blue). More than ten images were analysed in more than three different experiments. Scale bar = 1 μm. **G** 3D-SIM image showing location of EB1 (green) and ARK2 (purple) in a gametocyte activated for 3–4 min. DNA is stained with Hoechst dye (blue). More than ten images were analysed in more than three different experiments. Scale bar = 1 μm. **H** 3D-SIM image showing location of EB1 (purple) and cytoplasmic SAS4 (green) in gametocyte activated for 1 min. DNA is stained with Hoechst dye (blue). More than ten images were analysed in more than three different experiments. Scale bar = 1 μm. **I** STED confocal microscopy showing co-localisation of EB1 (purple) and α-tubulin (green) at spindle but not with cytoplasmic MTs in a gametocyte activated for 1 min. DNA is stained with SiR DNA (blue). More than ten images were analysed in more than three different experiments. Scale bar = 1 μm.

3D-SIM images of gametocytes expressing EB1-mCherry/SAS4-GFP; whereby SAS4 was located in the cytoplasm but aligned with the EB1 bridge in the nucleus (Fig. 6H and Supplementary Fig. 9A)[39]. We also performed STED microscopy on fixed EB1-GFP gametocytes stained with anti-GFP and anti-tubulin antibodies, which confirmed EB1's location on the spindle: the images showed EB1 on the nuclear spindle MTs with a distribution like that of ARK2 (Fig. 6I and Supplementary Fig. 8B).

### EB1-GFP is enriched at spindles and also associated with apical polarity during ookinete differentiation

Live-cell imaging of EB1-GFP during early meiosis in zygote differentiation located the protein on spindles and spindle poles, but it then disappeared as the ookinete matured, with a pattern like that observed for ARK2-GFP (Supplementary Fig. 10). In addition, EB1-GFP accumulated at the nascent apical end of most developing ookinetes, suggesting it is also important in defining cell polarity. In later stages of ookinete differentiation, EB1 was distributed around the periphery of the growing protuberance, but had disappeared in mature ookinetes (Supplementary Fig. 10A, B).

### EB1 is not essential for asexual blood-stage proliferation, but like ARK2 its deletion affects endomitosis during sporogony

We showed that ARK2 and EB1 have similar spatiotemporal dynamics but wanted to establish whether deletion of the EB1 gene would have a similar phenotype to that of an ARK2 mutant line. We therefore generated an EB1 gene-deletion mutant (Δeb1) via double homologous recombination (Supplementary Fig. 11A, B). This deletion had no effect on asexual blood-stage parasite development, in contrast to ARK2 that is essential during blood-stage schizogony[20]. Male gamete formation (exflagellation), fertilisation and zygote differentiation (ookinete development) were also unaffected in Δeb1 parasites (Fig. 7A, B). However, deletion of eb1 resulted in significantly reduced oocyst numbers on day 10 post infection of mosquitoes (Fig. 7C). The oocysts that were present were smaller than those of WT parasites (Fig. 7D), and by day 21 no oocysts were detectable (Fig. 7C, D) suggesting that development was completely blocked at some point beyond day 10.

Finally, to determine the global pattern of transcription in Δeb1 gametocytes, we performed RNA-seq analysis 30 min after activation. This analysis revealed that, in addition to the complete absence of EB1 transcripts (Fig. 7E), 129 and 411 genes were significantly downregulated and upregulated respectively at adjusted P value cut-off of 0.05 and $\log_2$fold change cut-off of 0.5 (Fig. 7F and Supplementary Data 4). RNA-seq genome browser tracks for Integrative Genomics Viewer (IGV) can be found at github.com/Sabel14/EB1_RNAseq_tracks, including tracks for 0 min after activation although these were not used for RNA-seq. GO enrichment analysis of upregulated genes using

R package TopGO with the weight01 algorithm for the Fisher Exact Test identified proteins involved in phosphorylation, transcription, and MT movement (Supplementary Fig. 11C).

### Reciprocal immunoprecipitation of EB1-GFP identifies its association with ARK2 at the spindle-kinetochore interface

To further analyse the association between ARK2 and EB1, we performed the reciprocal immunoprecipitation of EB1-GFP, from gametocytes 1-min post activation (Supplementary Fig. 12 and Supplementary Data 3). Comparison of the identified proteins (EB1-GFP versus WT-GFP) revealed a pattern of putative interactions for EB1 very similar to that of ARK2, but with much higher peptide support, in particular for both myosin K and MISFIT (Supplementary Fig. 12). Other peptides were derived from components of the kinetochore including INCENP2, ARK1, Stu2 and AKiT1[KNL1], suggesting that EB1 is located at the spindle-kinetochore interface like ARK2. Lastly, we found a strong enrichment of SMC proteins in comparison to WT-GFP or ARK2-GFP precipitates, including condensin (SMC2/4) and cohesin (SMC1/3) components, as well as proteins involved in DNA replication (MCM, ORC and RFC). To capture the quantitative similarity of proteins between GFP, EB1 and ARK2 experiments, we performed a PCA of the combined datasets of unique peptide spectral counts per protein for each pulldown (Supplementary Data 3). Using the ln(x)+1 transformed peptide values (with non-detected peptide values set to 0), our PCA captured 88.5 % of the variation within the data in the first two principal components (for other principal components, see Supplementary Data 3). ARK2, MISFIT, EB1 and MyoK clustered together, consistent with their proposed close association (Fig. 8A), and a similar pattern was observed for ARK2 and EB1 with the pre-replication complex component CDC6 (PBANKA_1102900). Furthermore, we observed co-variation for Mad1 (AKiT7), kinesin-8X, the RFC-like protein (PBANKA_0202500) and two polymerase subunits (Fig. 8A). Overall, we found co-variation in the 2 dimensional PCA projection of the ARK2/EB1-GFP pulldown data for proteins that are likely part of the same cellular structures or complexes, such as the kinetochore, spindle and various complexes involved in DNA replication (MCM/RFC/ORC), cohesin (SMC1/3), condensin (SMC2/4) and tubulin (alpha and beta together) (Fig. 8B), providing further confidence for the value of PCA to detect protein associations, possibly within the same complex.

In conclusion, results obtained from our proteomics experiments, in combination with functional analyses, suggest that the highly divergent Aurora kinase paralogue ARK2 and plasmodial lateral MT-binding protein, EB1 are part of a larger group of proteins including MyoK and MISFIT that are associated with the spindle apparatus, and that both EB1 and ARK2 are involved in the rapid cycles of spindle assembly and chromosome segregation during male gametony.

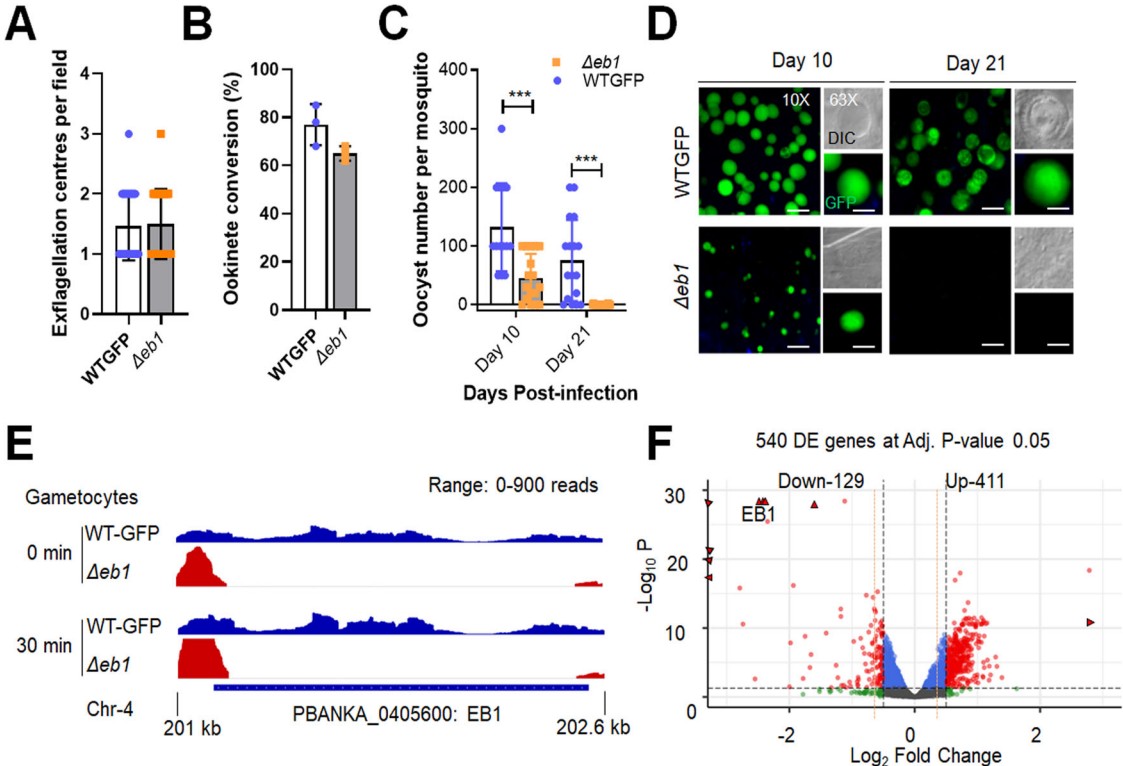

**Fig. 7 | Deletion of *Pbeb1* identifies an essential role in oocyst development and sporogony. A** The number of exflagellation centres per field of *Δeb1* (black bar) compared with WT-GFP (white bar) parasites at the end of male gametogony. Shown is mean ± SD *n* = 3 independent experiments. **B** Percentage ookinete conversion for *Δeb1* (black bar) and WT-GFP (white bar) parasites. Ookinetes were identified by reactivity with 13.1 antibody and successful differentiation into elongated 'banana-shaped' ookinetes. Shown is mean ± SD; *n* = 3 independent experiments. **C** Total number of GFP-positive oocysts per infected mosquito in *Δeb1* (black bar) and WT-GFP (white bar) parasites at 10- and 21 days post infection (dpi). Shown is mean ± SD; *n* = 3 independent experiments (with >15 mosquitoes for

each). Multiple comparison *t* test (non-parametric) showed significant differences in oocyst number ***$P < 0.001$. (*P* values: 0.000328−Day 10; 0.000016−Day 21). **D** Midguts at ×10- and ×63 magnification showing fluorescent oocysts of *Δeb1* and WT-GFP lines at 10 and 21 dpi. More than 20 images were analysed in more than three different experiments. Scale bar = 50 μm (×10) or 20 μm (×63). **E** RNA-seq analysis showing depletion of EB1 transcript in *Δeb1* gametocytes at 0- and 30 min post activation. Data range is 0 to 900 reads. **F** Scatter plot showing up- and downregulated genes in *Δeb1* compared to WT-GFP gametocytes, with adjusted *P* value cut-off of 0.05. Source data are provided as a Source Data file.

## Discussion

Aurora is a serine-threonine kinase family that is highly conserved in eukaryotes. Previous phylogenetic analyses had shown that the family evolved from a single ancestral kinase by widespread recurrent duplications throughout eukaryotic evolution[3]. AKs play crucial roles in mitotic/meiotic entry, bipolar spindle assembly, chromosome segregation and cytokinesis; and work in conjunction with scaffold proteins like chromosome passenger protein (CPC)[3,14,40]. The three divergent *Plasmodium* ARKs are essential for asexual parasite proliferation in the mammalian host but their role in sexual stages in the mosquito and the presence or absence of scaffold proteins were unknown[19,20]. Here we focus on the location and function of *P. berghei* ARK2, an Aurora kinase-related protein and the unique plasmodial lateral MT-binding protein (EB1) during the unconventional modes of cell proliferation, differentiation and division during endomitosis and meiosis of the sexual transmission stages within the mosquito host.

Our in-depth bioinformatics analysis confirmed the presence of three divergent ARKs and the absence of many scaffold proteins, corroborating earlier studies showing that *Plasmodium* lacks scaffold components like survivin and borealin[41]. However, as reported for *Toxoplasma*[12] two members of INCENP are present. In human cells Aurora A associates with spindle MTs and centrosomes, while Aurora B is located at centromeres, the spindle and the midbody[4,9,42]. In other eukaryotes, similar patterns of sub-functionalisation are found (Supplementary Fig. 1), with one paralogue termed the equatorial AK (Aurora B in humans) and the other the polar AK (Aurora A in humans).

Although it is difficult to assign the conserved AK homologue by similar subcellular location of *Plasmodium* ARK2, it appears that ARK2 is more like Aurora A due to its association with spindles and with the acentriolar inner MTOC. If correct, this analysis would suggest that ARK2 is the polar AK. Of the other *Plasmodium* ARKs, we predict that ARK1 is likely the equatorial ARK due to its conventional length (Supplementary Fig. 1) and the association of the *T. gondii* ARK1 orthologue with INCENP1-2[12]. The presence of a third AK in *Plasmodium* suggests either an additional sub-functionalisation of the canonical equatorial/polar AKs or a new function that may have been adopted by ARK3. For both ARK1 and ARK3, similar experiments to those performed here are needed to reveal their interactions and functions.

Using live-cell imaging, we show a very discrete and dynamic pattern of ARK2 location during mitosis in male gametogony. The protein transitions from a diffuse nuclear distribution before gametocyte activation to a location at the spindle poles and then moves with the spindle during the three mitotic cycles. A similar pattern is also observed during the first meiotic stages in the developing ookinete. There is no clear anaphase observed in these cells. Live-cell imaging of dual-fluorescent lines expressing ARK2-GFP and either the kinetochore marker NDC80-mCherry or basal body marker Kinesin-8B-mCherry demonstrates that ARK2 occupies a unique location, associated with both spindle MTs and the kinetochore during spindle formation, and is located at the spindle pole of the inner acentriolar MTOC, but not at the cytoplasmic centriolar MTOC that includes the basal body marker SAS4 or Kinesin-8B[39,43]. STED microscopy with alpha-tubulin and

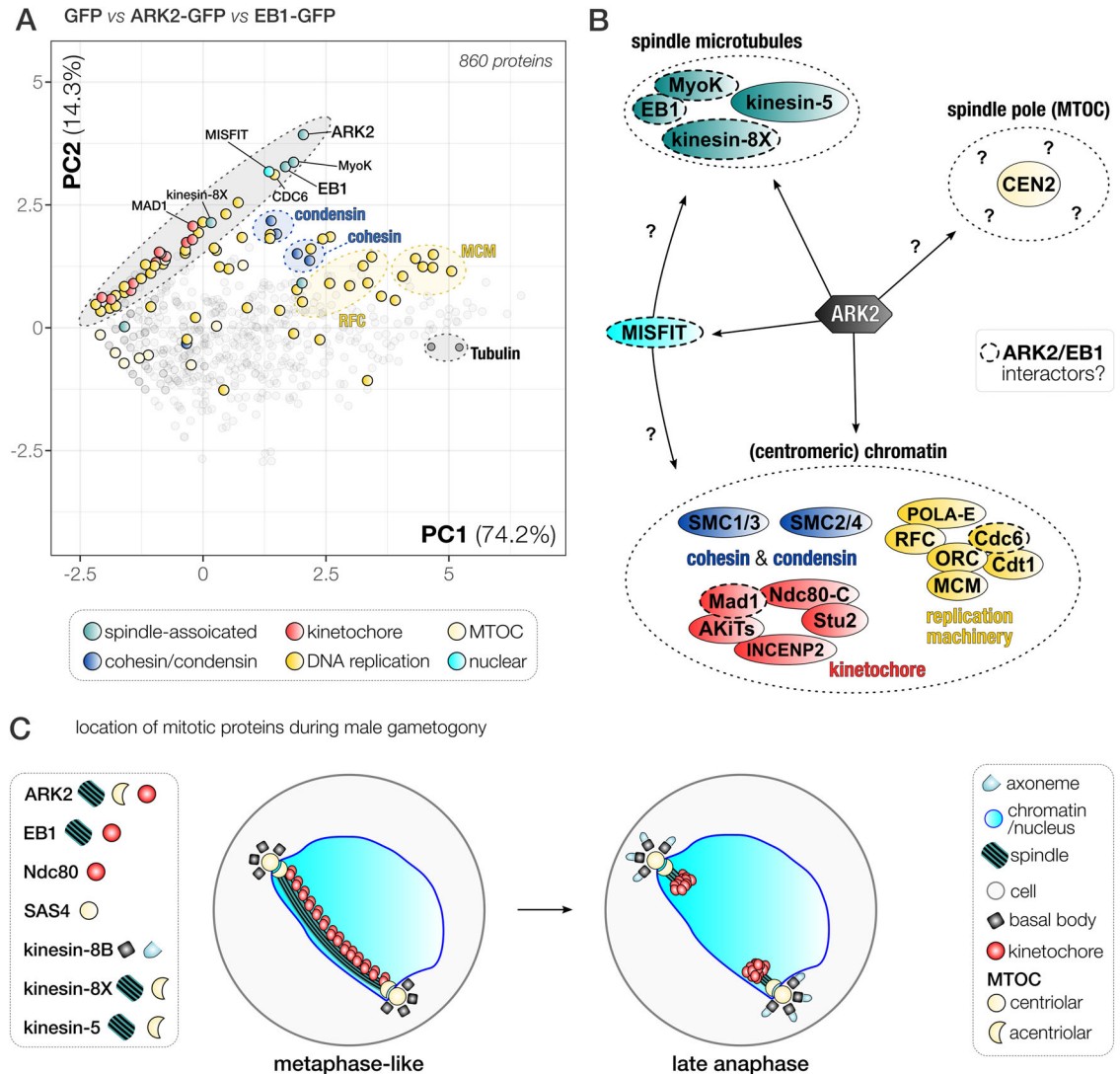

**Fig. 8 | EB1 and ARK2 are part of a network of spindle-associated proteins during male gametogony. A** Projection of the first two components (85.5% variation) of a principal component analysis based on log-transformed unique peptide values as identified by mass spectrometry from ARK2-GFP, EB1-GFP and WT-GFP immunoprecipitates. Clusters of proteins that we identified indicate physical and/ or functional association, e.g., the MCM (helicase), RFC (replication factor C), condensin (SMC2/4), cohesion (SMC1/3), pre-RC/ORC complex, tubulin (alpha-beta subunit) and parts of the kinetochore. EB1 and ARK2 are part of a larger network of spindle, kinetochore and DNA replication protein (black-shaded ellipsoid), which includes among others MAD1, kinesin-8X, CDC6, MIFSFIT and MyoK. **B** Schematic that reconciles the ARK2 location relative to that of other relevant proteins present at the spindle (pole) during male gametogony. Proteins are grouped by cellular structure/complexes (see legend in **A**). A dashed line indicates putative interactors of ARK2 and EB1. **C** Location of different mitotic proteins during male gametogony. Shown are two phases of mitosis: a metaphase-like state (left) with kinetochores populating the full length of the spindle, and late anaphase (right).

super-resolution images of dual fluorescence-tagged lines further corroborate this unusual location of ARK2. The number of acentriolar MTOCs, defined by the location of ARK2-GFP during male gamete formation and zygote differentiation correlates with the ploidy of the cell, for example, in the 2N, 4N or 8N male cell there were two, four or eight foci, and in the 4N ookinete there were four foci. Interestingly the ARK2-GFP signal disappeared by 12 h of zygote development, when it is likely that the second meiotic division had taken place (without karyokinesis), whereas NDC80 was present until the end of the ookinete stage but in both cases four fluorescent foci are seen in the 4N ookinete.

EB1 is a unique lateral MT-tracking protein that is involved in spindle dynamics in *Plasmodium* as shown here and in a recent report[44]. During male gametogony, EB1 had a location like that of ARK2 on the spindle and acentriolar MTOC. The parasite line expressing dual-fluorescent-tagged EB1 and ARK2 showed that they are associated with each other at the spindle during different stages of development

in both male gamete formation and zygote differentiation. STED imaging of EB1 showed that EB1 is associated with the spindle as also observed for ARK2. A recent study further confirmed that EB1 binds to full length spindle MTs and regulates this spindle structure[44]. Intriguingly, EB1 was not detected in the proliferative asexual stages within red blood cells. This is in contrast to the presence of EB1 during the non-mitotic gametocytogenesis in *P. falciparum*[45] and in asexual cell proliferation in *Toxoplasma* where EB1 was observed associated with spindle MTs[46]. These findings suggest that both ARK2 and EB1 are specifically associated with the spindle machinery and the acentriolar MTOC during male gamete formation and ookinete development. Furthermore, it has been revealed that *Plasmodium* EB1 has lost its MT plus end feature but possess MT lattice affinity, possibly because it is an orthologue distinct from canonical eukaryote EB1[44].

Our previous *Plasmodium* kinome screen showed that ARK2 has an essential role during blood-stage development[20]. Here we used conditional knockdown approaches to study the function of ARK2 in

sexual stages, and our results show that our *ark2* and *eb1* knockdown mutants have a similar phenotype to those of *Plasmodium*-specific cyclin, PbCYC3, and kinesin-8X, in which oocyst size and sporozoite formation were affected[31,47], and similar to what is observed in other deletion mutants including MISFIT, PK7 and PPM5 genes[20,30,37]. Genetic backcross experiments with *dozi* and *hap2* mutants that affect female and male gametogony, respectively, demonstrated that the $P_{clag}$.*ark2* defect in oocyst development is inherited as a defect in the male gametocyte lineage, similar to what is observed for Δ*misfit* and Δ*ppm5*, for which there is an absolute requirement for a functional gene from the male line[30,37]. In each case, the genetic defect is transferred through the male lineage and manifest during endomitosis in the oocyst; thereby blocking parasite development and transmission.

Similarly, our data suggest that both ARK2 and EB1 are part of the spindle assembly, and although male gametes and ookinetes are produced, downregulated ARK2 expression or deletion of *eb1* have a delayed effect contributed through the male lineage that is seen during oocyst development and results in a complete block in parasite transmission.

Global transcript analysis showed significant differences in gene expression between the knockdown $P_{clag}$.*ark2* and WT parasite lines. Genes coding for proteins involved in MT-based movement and regulation of gene expression were mostly affected, including a large number of protein kinases; several motor proteins (e.g., kinesin and dynein); and proteins involved in invasion or oocyst development. This finding is consistent with global phosphoproteomic studies of male gametogony, in which ARK2 was shown to be associated with rapid phosphorylation of MT proteins in either very early or late stages of male gamete formation[48]. It is possible that ARK2 phosphorylates various substrates, including kinesin-8X, EB1, and MISFIT.

Our results suggest that ARK2 is largely located on the spindle apparatus, suggesting that it is not part of a CPC-like complex. We confirmed the earlier phylogenetic studies that showed that CPC components like Survivin and Borealin are absent and ARK2-GFP immunoprecipitations identified unique candidate ARK2-associated proteins that are part of kinetochore components and proteins with a role at the spindle apparatus. These proteins included the EB1, the myosin MyoK, a nuclear formin-like protein called MISFIT, members of the NDC80 outer kinetochore complex, and other Apicomplexan Kinetochore proteins (AKiTs). The presence of such proteins strongly suggests that ARK2 associates with the kinetochore-spindle MT interface. A reciprocal pulldown with EB1-GFP identified a similar set of proteins as found in the ARK2-GFP pulldown experiment. Components of the kinetochore like MAD1, NDC80, STU2 and AkiT were detected although no high abundance proteins were present in either ARK2 or EB1 immunoprecipitates. In addition, none of the TPX2 complex components were detected, suggesting that ARK2 is not functionally similar to Aurora A (polar Aurora kinase) of model eukaryotes[3]. How ARK2 associates with the spindle and whether EB1/MISFIT/MyoK and/or kinesin-8X play roles have yet to be determined by further biochemical experiments.

*Plasmodium* has only one EB1 homologue, compared to the three EB1 proteins present in other eukaryotes[49]. EB1 function is heavily regulated by protein phosphorylation: a cluster of six serine residues present in the linker region of the yeast EB1 homologue (Bim1) is phosphorylated by the AK homologue lpl1, regulating disassembly of the spindle midzone during anaphase. Human EB1 is co-immunoprecipitated with Aurora B[50]; the EB1 concentrates Aurora B at inner centromeres in a MT-dependent manner, resulting in phosphorylation of both kinetochore and chromatin substrates[51]. The likely association of ARK2 with spindle factors like EB1, MyoK and MISFIT was revealed through co-variation of these proteins in ARK2 and EB1-GFP immunoprecipitates. These data suggest that *Plasmodium* ARK2 may form a unique complex that has not been described in other organisms, but more experimental work is needed to test this hypothesis.

MyoK has been shown to be involved in mitosis in many organisms, however MISFIT is a *Plasmodium*-specific formin[37]. How ARK2 might interact with EB1, MyoK and MISFIT is unclear. We did not detect any conserved features amongst apicomplexan ARK2 orthologues apart from their kinase domain. Furthermore, structural prediction by AlphaFold2 (Supplementary Fig. S1) suggests that PbARK2 mostly consists of random coil. Interactions might thus be mediated by short linear motifs or non-specific interactions through patches of charged residues possibly facilitating phase separation near spindle microtubules. However, it is unknown whether MyoK and MISFIT are part of the spindle, and this will be investigated in future studies.

Overall, this study suggests that *Plasmodium* ARK2 is an Aurora kinase that is located at the spindle apparatus. It associates with EB1 and some other kinetochore-binding proteins, but in a dissimilar way to Aurora paralogues in model eukaryotes, such as Aurora B, which is CPC based (INCENP/Borealin/Survivin), or Aurora A (TPX/Cep192/BORA). Hence ARK2 is a unique Aurora paralogue that may associate with the spindle through interactions with EB1/MISFIT/MyoK. This model underlines the flexibility of molecular networks to rewire and drive unconventional modes of spindle organisation and chromosome segregation during cell division in the malaria parasite *Plasmodium*.

## Methods

### Ethics statement and mice details

The animal work passed an ethical review process and was approved by the United Kingdom Home Office. Work was carried out under UK Home Office Project Licenses (30/3248 and PDD2D5182) in accordance with the UK 'Animals (Scientific Procedures) Act 1986'. Six- to eight-week-old female CD1 outbred mice from Charles River laboratories were used for all experiments. The conditions of mice kept were a 12 h light and 12 h dark (7 am till 7 pm) light cycle, the room temperature was kept between 20 and 24 °C and the humidity was kept between 40 and 60%.

### Generation of transgenic parasites and genotype analyses

To generate the GFP-tag lines, a region of each gene (*ark2 and eb1*) downstream of the ATG start codon was amplified, ligated to p277 vector, and transfected as described previously[52]. The p277 vector contains the human *dhfr* cassette, conveying resistance to pyrimethamine. A schematic representation of the endogenous gene locus, the constructs and the recombined gene locus can be found in Supplementary Figs. 2A and 8A. For the parasites expressing the C-terminal GFP-tagged protein, diagnostic PCR was used with primer 1 (Int primer) and primer 3 (ol492) to confirm integration of the GFP targeting construct (Supplementary Figs. 2B and 8B). A list of primers used to amplify these genes can be found in Supplementary Data 5.

For the generation of the transgenic *ark2*-AID/HA line, library clone PbG01-2471h08 from the PlasmoGEM repository (http://plasmogem.sanger.ac.uk/) was used. Sequential recombineering and gateway (GW) steps were performed as previously described[53,54]. Insertion of the GW cassette following gateway reaction was confirmed using primer pairs GW1 (CATACTAGCCATTTTATGTG) × *ark2* QCR1 (GCTTTGCAGCCG AAGCTCCG) and GW2 (CTTTGGTGACAGATACTAC) × *ark2* QCR2 (AGGGGGAAAATGTTACACATGCGT). The modified library inserts were then released from the plasmid backbone using *Not*I. The *ark2*-AID/HA targeting vector was transfected into the 615-parasite line, and conditional degradation of ARK-AID/HA in the non-clonal line was performed, as described previously[35]. A schematic representation of the endogenous *ark2* locus, the constructs and the recombined *ark2* locus can be found in Supplementary Fig. 6A. A diagnostic PCR was performed for *ark2* gene knockdown parasites as outlined in Supplementary Fig. 6A. Primer pairs *ark2* QCR1/GW1, and *ark2* QCR2/GW2 were used to determine successful integration of the targeting construct at the 3' end of the gene (Supplementary Fig. 6B).

The conditional knockdown construct $P_{clag}.ark2$ was derived from $P_{clag}$ (pSS367) by placing *ark2* under the control of the *clag* gene (PBANKA_083630) promoter, as described previously[33]. A schematic representation of the endogenous *ark2* locus, the constructs and the recombined *ark2* locus can be found in Supplementary Fig. 6E. A diagnostic PCR was performed for *ark2* gene knockdown parasites as outlined in Supplementary Fig. 6E. Primer 1 (5'-intPTD24) and Primer 2 (5'-intPTD) were used to determine successful integration of the targeting construct at the 5' end of the gene. Primer 3 (3'-intPTclag) and Primer 4 (3'-intPTD24) were used to determine successful integration for the 3' end of the gene locus (Supplementary Fig. 6F). All the primer sequences can be found in Supplementary Data 5.

To study the function of EB1, the gene-deletion targeting vector for *eb1 was* constructed using the pBS-DHFR plasmid, which contains polylinker sites flanking a *T. gondii dhfr/ts* expression cassette conferring resistance to pyrimethamine, as described previously[20]. The 5' upstream sequence of *eb1* was amplified from genomic DNA and inserted into *Apa*I and *Hin*dIII restriction sites upstream of the *dhfr/ts* cassette of pBS-DHFR. A DNA fragment amplified from the 3' flanking region of *eb1* was then inserted downstream of the *dhfr/ts* cassette using *Eco*RI and *Xba*I restriction sites. The linear targeting sequence was released using *Apa*I/*Xba*I. A schematic representation of the endogenous *eb1* locus, the construct and the recombined *eb1* locus can be found in Supplementary Fig. 8A. The primers used to generate the mutant parasite lines can be found in Supplementary Data 5. A diagnostic PCR was used with primer 1 (IntN138_5) and primer 2 (ol248) to confirm integration of the targeting construct, and primer 3 (KO1) and primer 4 (KO2) were used to confirm deletion of the *eb1* gene (Supplementary Fig. 8B and Supplementary Data 5). *P. berghei* ANKA line 2.34 (for GFP-tagging) or ANKA line 507cl1 expressing GFP (for the gene-deletion and knockdown construct) parasites were transfected by electroporation[55].

### Live-cell imaging

To examine ARK2-GFP and EB1-GFP expression during erythrocytic stages, parasites growing in schizont culture medium were used for imaging at different stages of schizogony. Purified gametocytes were examined for GFP expression and cellular location at different time points (0, 1–15 min) after activation in ookinete medium[47]. Zygote and ookinete stages were analysed throughout 24 h of culture using cy3-conjugated 13.1 antibody (red), which recognises P28 protein on the surface of zygotes and ookinetes. Oocysts and sporozoites were imaged using infected mosquito guts. Images were captured using a ×63 oil immersion objective on a Zeiss AxioImager M2 microscope fitted with an AxioCam ICc1 digital camera.

### Generation of dual-tagged parasite lines and analysis of colocalization

The green (GFP)- or red (mCherry)-tagged ARK2 and EB1 parasite lines were mixed with mCherry- or GFP-tagged lines of kinetochore marker NDC80[29], axoneme marker kinesin-8B[56] and basal body marker SAS4[39] in equal numbers and injected into mice. Mosquitoes were fed on these mice 4 to 5 days after infection when gametocytemia was high, and were checked for oocyst development and sporozoite formation at day 14 and day 21 after feeding. Infected mosquitoes were then allowed to feed on naïve mice and after 4 to 5 days the mice were examined for blood-stage parasitaemia by microscopy with Giemsa-stained blood smears. Some parasites expressed both ARK2-mCherry and NDC80-GFP; and ARK2-GFP and kinesin-8B-cherry in the resultant gametocytes, and these were purified, and fluorescence microscopy images were collected as described above.

Colocalization of green and red fluorescence was quantified as described previously[57] and Pearson's colocalization coefficient (R) was calculated using image J software (version 1.44).

### Parasite phenotype analyses

Blood samples containing approximately 50,000 parasites of the ark2-knockdown/*eb1* knockout lines were injected intraperitoneally (i.p.) into mice to initiate infection. Asexual stages and gametocyte production were monitored by microscopy on Giemsa-stained thin smears. Four to 5 days post infection, exflagellation and ookinete conversion were examined as described previously[52] with a Zeiss AxioImager M2 microscope (Carl Zeiss, Inc.) fitted with an AxioCam ICc1 digital camera. To analyse mosquito infection and transmission, 30 to 50 *Anopheles stephensi* SD 500 mosquitoes were allowed to feed for 20 min on anaesthetised, infected mice with at least 15% asexual parasitaemia and carrying comparable numbers of gametocytes as determined on Giemsa-stained blood films. To assess midgut infection, ~15 guts were dissected from mosquitoes on days 7 and 14 post feeding and oocysts were counted using a ×63 oil immersion objective. On day 21 post feeding, another 20 mosquitoes were dissected, and their guts and salivary glands crushed separately in a loosely fitting homogeniser to release sporozoites, which were then quantified using a haemocytometer or used for imaging. Mosquito bite-back experiments were performed 21 days post feeding using naive mice, and blood smears were examined after 3–4 days.

### Purification of gametocytes

The purification of gametocytes was achieved by injecting parasites into phenylhydrazine-treated mice[58] and gametocyte enrichment by sulfadiazine treatment after 2 days of infection. The blood was collected on day 4 after infection, and gametocyte-infected cells were purified on a 48% v/v NycoDenz (in PBS) gradient (NycoDenz stock solution: 27.6% w/v NycoDenz in 5 mM Tris-HCl, pH 7.20, 3 mM KCl, 0.3 mM EDTA). The gametocytes were harvested from the interface and activated.

### Immunoprecipitation and mass spectrometry

Male gametocytes of ARK2-GFP and EB1-GFP parasites were used at 1 min post activation to prepare cell lysates. WT-GFP gametocytes were used as controls. Purified parasite pellets were cross-linked using formaldehyde (10 min incubation with 1% formaldehyde, followed by 5 min incubation in 0.125 M glycine solution and three washes with phosphate-buffered saline (PBS; pH 7.5). Immunoprecipitation was performed using the protein lysates and a GFP-Trap_A Kit (Chromotek) following the manufacturer's instructions. Briefly, the lysates were incubated for 2 h with GFP-Trap_A beads at 4 °C with continuous rotation. Unbound proteins were washed away, and proteins bound to the GFP-Trap_A beads were digested using trypsin. The tryptic peptides were analysed by liquid chromatography–tandem mass spectrometry. Mascot (http://www.matrixscience.com/) and MaxQuant (https://www.maxquant.org/) search engines were used for mass spectrometry data analysis. Experiments were performed in duplicate. Peptide and proteins having a minimum threshold of 95% were used for further proteomic analysis. The PlasmoDB database was used for protein annotation, and a separate manual curation was performed to classify proteins into five categories relevant for functional annotation of ARK2/EB1 immunoprecipitates: cohesin/condensin, DNA repair/replication, kinetochore, MTOC, spindle, proteasome and ribosome/translation. The latter two categories were not further used in our analyses, but these are often found in the background of GFP-pulldowns in *Plasmodium* (Supplementary Data 3). To capture co-variation of pulldown proteins between different experiments (GFP-only versus ARK2-GFP versus EB1-GFP) we performed principal component analysis (PCA) using unique peptide values per protein present in each of the samples. Values for undetected proteins were set to 0. Values were $\ln(x + 1)$ transformed and PCA was performed using the ClustVis webserver (settings Nipals PCA, no scaling)[59]. The first six principal components for the analysis comparing ARK2/EB1/GFP-only samples can be found in Supplementary Data 3.

## Evolutionary bioinformatics

Phyletic profiles were derived from previous studies[41,60,61]. Sequence searches for missing homologues (Mps1, Bub1, Polo, Survivin, Borealin) were performed with HHsearch[62] and the hmmer package[63], as previously described[29]. Additional similarity searches for TPX2, Bora, Cep192 were executed as published previously[41]. Uniprot Identifiers and common gene names for all sequences in this study can be found in Supplementary Data 1.

## Ookinete motility assays

The motility of $P_{clag}.ark2$ ookinetes was assessed using Matrigel as described previously[64,65]. Ookinete cultures grown for 24 h were added to an equal volume of Matrigel (Corning), mixed thoroughly, dropped onto a slide, covered with a coverslip, and sealed with nail polish. The Matrigel was then allowed to set at 20 °C for 30 min. After identifying a field containing an ookinete, time-lapse videos (one frame every 5 s for 100 cycles) were collected using the differential interference contrast settings with a ×63 objective lens on a Zeiss AxioImager M2 microscope fitted with an AxioCam ICc1 digital camera and analysed with the AxioVision 4.8.2 software.

## Fixed immunofluorescence assay and deconvolution microscopy

The ARK2-GFP gametocytes were purified, activated in ookinete medium, fixed at different time points with 4% paraformaldehyde (PFA, Sigma) diluted in MT-stabilising buffer (MTSB) for 10–15 min, and added to poly-L-lysine coated slides. Immunocytochemistry was performed using primary GFP-specific rabbit monoclonal antibody (mAb) (Invitrogen-A1122; used at 1:250) and primary mouse anti-α tubulin mAb (Sigma-T9026; used at 1:1000). Secondary antibodies were Alexa 488 conjugated anti-mouse IgG (Invitrogen-A11004) and Alexa 568 conjugated anti-rabbit IgG (Invitrogen-A11034) (used at 1 in 1000). The slides were then mounted in Vectashield with DAPI (Vector Labs) for fluorescence microscopy. Parasites were visualised on a Zeiss AxioImager M2 microscope fitted with an AxioCam ICc1 digital camera. Post-acquisition analysis was carried out using Icy software−version 1.9.10.0. Images presented are 2D projections of deconvoluted Z-stacks of 0.3 μm optical sections.

## STED microscopy

Immunofluorescence staining for STED microscopy was performed as a combination of protocols described previously[66,67]. Briefly, the gametocytes were fixed with 4% pre-warmed PFA/PBS. PFA was washed away thrice with PBS. Fixed cells were stored in PBS at 4 °C in the dark for later immunofluorescence staining. Before beginning the immunofluorescence procedure, glass-coated 35 mm imaging μ-dishes (Ibidi, 81156) were coated with poly-L-lysine (PLL) solution (0.01%, Sigma-Aldrich, P4832) according to the manufacturer's guidelines (0.05% final solution). After extensive washing with nuclease-free water, dishes were left to dry. The fixed cells in PBS were then seeded into PLL-coated dishes and left to settle for a day. The cells were then washed with PBS, permeabilized with 0.5% Triton X-100/PBS for 30 min at room temperature and rinsed three times with PBS. To quench free aldehyde groups, cells were incubated with freshly prepared 0.1 mg/ml NaBH₄/PBS solution for 10 min. Cells were rinsed thrice with PBS and blocked with 3% BSA/PBS for 30 min. In the meantime, primary antibodies were diluted in 3% BSA/PBS and the solution was centrifuged at 21,100× $g$ for 10 min at 4 °C to remove potential aggregates. Cells were incubated with primary antibody to stain tubulin (mouse anti-α-tubulin B-5-1–2 mAb, Sigma-Aldrich, T5168, dilution 1:250) for 4 h at room temperature. Next, the cells were washed three times with 0.5% Tween-20/PBS. Incubation with secondary antibodies (donkey anti-mouse IgG Alexa Fluor 594, Abcam, ab150112; RRID: AB_2813898, dilution 1:500 or STAR ORANGE, goat anti-mouse IgG, Abberior GmbH, STORANGE-1001-

500UG, dilution 1:500) in 3% BSA/PBS was performed for 1 h after removal of aggregates as described for primary antibodies. After washing twice with 0.5% Tween-20/PBS and once with PBS, cells were incubated with SiR-DNA solution (Spirochrome, SC007, 1:100) for 1 h, then washed once washed with PBS and stored in PBS at 4 °C in the dark until imaging.

Rescue-STED microscopy was performed on a single-point scanning Expert Line easy3D STED super-resolution microscope (Abberior Instruments GmbH), equipped with a pulsed 775 nm STED depletion laser and two avalanche photodiodes for detection. Super-resolution images were acquired with a 100×1.4 NA objective, a pixel size of 10–20 nm and a pixel dwell time of 8 μs. The STED laser power was set to 15–30%, whereas the other lasers (488, 594 and 640 nm) were adjusted to the antibody combinations used. To acquire z-stacks, a total z-stack of 3–5 μm was acquired using a z-step size of 200–300 nm. The channel where the 640 nm laser was used for SiR-DNA excitation was taken separately in time but in the same imaging region as used for the 488 and 594 channels, and with custom-made defined emission boundaries of 594 and 640 to limit signal crosstalk between the channels. 640 and 594 nm channels were taken with STED depletion laser using the parameters described above, whereas 488 channel was taken with the same parameters but without STED depletion laser. STED images were assembled in Fiji (Image J-win64) as maximum intensity projections of acquired z-stacks that contained noticeable EB1 and ARK2 signals.

## Ultrastructure expansion microscopy (UExM)

Purified gametocytes were activated for 1–2 min and then activation was stopped by adding 4% formaldehyde. Sample preparation of *P. berghei* parasites for UExM was performed as previously described[68,69], except that 4% formaldehyde (FA) was used as fixative[43]. Fixed samples were then attached on a 12 mm round poly-D-lysine (A3890401, Gibco) coated coverslip for 10 min. Immuno-labelling was performed using primary antibodies against α-tubulin and β-tubulin (1:200 dilution, AA344 and AA345 from the Geneva antibody facility), anti-γ-tubulin antibody (1:500 dilution, Sigma T5192) and anti-HA antibody (3F10) (1:250 dilution, Roche). Secondary antibodies anti-guinea pig Alexa 647, anti-rabbit Alexa 405 and anti-rat Alexa 488 were used at dilutions 1:400 (Invitrogen). Atto 594 NHS-ester was used for bulk proteome labelling (Merck 08741). Images were acquired on a Leica TCS SP8 microscope, image analysis was performed using Fiji-Image J and Leica Application Suite X (LAS X) software.

## Structured illumination microscopy

A small volume (3 μl) of gametocytes was mixed with Hoechst dye and pipetted onto 2% agarose pads (5 × 5 mm squares) at room temperature. After 3 min, these agarose pads were placed onto glass bottom dishes with the cells facing towards glass surface (MatTek, P35G-1.5-20-C). Alternatively, 2 μl of cell suspension was placed on microscope slide and covered with long (50x34mm) coverslip, to obtain a very thin monolayer and to immobilise the cells. Cells were scanned with an inverted microscope using Zeiss Plan-Apochromat 63x/1.4 Oil immersion or Zeiss C-Apochromat 63×/1.2 W Korr M27 water immersion objective on a Zeiss Elyra PS.1 microscope, using the structured illumination microscopy (SIM) technique. The correction collar of the objective was set to 0.17 for optimum contrast. The following settings were used in SIM mode: lasers, 405 nm: 20%, 488 nm: 16%, 561 nm: 8%; exposure times 200 ms (Hoechst) 100 ms (GFP) and 200 ms (mCherry); three grid rotations, five phases. The bandpass filters BP 420-480 + LP 750, BP 495-550 + LP 750 and BP 570-620 + 750 were used for the blue, green and red channels, respectively. Multiple focal planes (Z-stacks) were recorded with 0.2 μm step size; later post-processing, a Z correction was done digitally on the 3D rendered images to reduce the effect of spherical aberration (reducing the elongated view in Z; a process previously tested with 0.1 μm fluorescent beads,

ThermoFisher T7284). Registration correction was applied based on control images of multicolour fluorescent beads. Images were processed and all focal planes were digitally merged into a single plane (Maximum intensity projection). The images recorded in multiple focal planes (Z-stack) were 3D rendered into virtual models and exported as images and movies (see supplementary material). Processing and export of images and videos were done by Zeiss Zen 2012 Black edition, Service Pack 5 and Zeiss Zen 2.1 Blue edition.

## RNA isolation and quantitative real-time PCR (qRT-PCR) analyses

RNA was isolated from purified gametocytes using an RNA purification kit (Stratagene). cDNA was synthesised using an RNA-to-cDNA kit (Applied Biosystems). Gene expression was quantified from 80 ng of total RNA using SYBR green fast master mix kit (Applied Biosystems). All the primers were designed using primer 3 (Primer-blast, NCBI). Analysis was conducted using an Applied Biosystems 7500 fast machine with the following cycling conditions: 95 °C for 20 s followed by 40 cycles of 95 °C for 3 s; 60 °C for 30 s. Three technical replicates and three biological replicates were performed for each assayed gene. The *hsp70* (PBANKA_081890) and *arginyl-t RNA synthetase* (PBANKA_143420) genes were used as endogenous control reference genes. The primers used for qPCR can be found in Supplementary Data 5.

## RNA-seq analysis

Libraries were prepared from lyophilised total RNA, first by isolating mRNA using NEBNext Poly(A) mRNA Magnetic Isolation Module (NEB), then using NEBNext Ultra Directional RNA Library Prep Kit (NEB) according to the manufacturer's instructions. Libraries were amplified for a total of 12 PCR cycles (12 cycles of [15 s at 98 °C, 30 s at 55 °C, 30 s at 62 °C]) using the KAPA HiFi HotStart Ready Mix (KAPA Biosystems). Libraries were sequenced using a NovaSeq 6000 DNA sequencer (Illumina), producing paired-end 100-bp reads.

FastQC (https://www.bioinformatics.babraham.ac.uk/projects/fastqc/) was used to analyse raw read quality. The first 11 bp of each read and any adaptor sequences were removed using Trimmomatic (http://www.usadellab.org/cms/?page=trimmomatic). Bases were trimmed from reads using Sickle with a Phred quality threshold of 25 (https://github.com/najoshi/sickle). The resulting reads were mapped against the *P. berghei* ANKA genome (v36) using HISAT2 (version 2-2.1.0), using default parameters. Uniquely mapped, properly paired reads with mapping quality 40 or higher were retained using SAMtools (http://samtools.sourceforge.net/). Genome browser tracks were generated and viewed using the Integrative Genomic Viewer (IGV) (Broad Institute). Raw read counts were determined for each gene in the *P. berghei* genome using BedTools (https://bedtools.readthedocs.io/en/latest/#) to intersect the aligned reads with the genome annotation. Differential expression analysis was done by use of R package DESeq2 to call up- and downregulated genes with an adjusted P-value cut-off of 0.05. Gene ontology enrichment was done using R package topGO (https://bioconductor.org/packages/release/bioc/html/topGO.html) with the weight01 algorithm.

## ChIP-seq analysis

Gametocytes of EB1-GFP and NDC80-GFP (as a positive control) parasites were harvested, and the pellets were resuspended in 500 µl of Hi-C lysis buffer (25 mM Tris-HCl, pH 8.0, 10 mM NaCl, 2 mM AESBF, 1% NP-40, protease inhibitors). After incubation for 10 min at room temperature (RT), the resuspended pellets were homogenised by passing through a 26.5 gauge needle/syringe 15 times and cross-linked by adding formaldehyde (1.25% final concentration) for 25 min at RT with continuous mixing. Cross-linking was stopped by adding glycine to a final concentration of 150 mM and incubating for 15 min at RT with continuous mixing. The sample was centrifuged for 5 min

at 2500×*g* (~5000 rpm) at 4 °C, the pellet washed once with 500 µl ice-cold wash buffer (50 mM Tris-HCl, pH 8.0, 50 mM NaCl, 1 mM EDTA, 2 mM AESBF, protease inhibitors) and the pellet stored at −80 °C for ChIP-seq analysis. The cross-linked parasite pellets were resuspended in 1 mL of nuclear extraction buffer (10 mM HEPES, 10 mM KCl, 0.1 mM EDTA, 0.1 mM EGTA, 1 mM DTT, 0.5 mM AEBSF, 1× protease inhibitor tablet), post 30 min incubation on ice, 0.25% Igepal-CA-630 was added and the sample homogenised by passing through a 26 G× ½ needle. The nuclear pellet extracted through 5000 rpm centrifugation, was resuspended in 130 µl of shearing buffer (0.1% SDS, 1 mM EDTA, 10 mM Tris-HCl pH 7.5, 1× protease inhibitor tablet), and transferred to a 130 µl Covaris sonication microtube. The sample was then sonicated using a Covaris S220 Ultrasonicator for 8 min (Duty cycle: 5%, intensity peak power: 140, cycles per burst: 200, bath temperature: 6 °C). The sample was transferred to ChIP dilution buffer (30 mM Tris-HCl pH 8.0, 3 mM EDTA, 0.1% SDS, 30 mM NaCl, 1.8% Triton X-100, 1× protease inhibitor tablet, 1× phosphatase inhibitor tablet) and centrifuged for 10 min at 13,000 rpm at 4 °C, retaining the supernatant. For each sample, 13 µl of protein A agarose/salmon sperm DNA beads were washed three times with 500 µl ChIP dilution buffer (without inhibitors) by centrifuging for 1 min at 1000 rpm at room temperature, then buffer was removed. For pre-clearing, the diluted chromatin samples were added to the beads and incubated for 1 h at 4 °C with rotation, then pelleted by centrifugation for 1 min at 1000 rpm. Before adding antibody, ~10% of one EB1-GFP sample was taken as input. The supernatant was removed into a LoBind tube, carefully so as not to remove any beads, and 2 µg of anti-GFP antibody (Abcam ab290, anti-rabbit) were added to the sample and incubated overnight at 4 °C with rotation. For one EB1-GFP sample, IgG antibody (ab37415) was added instead as a negative control. Per sample, 25 µl of protein A agarose/salmon sperm DNA beads were washed with ChIP dilution buffer (no inhibitors), blocked with 1 mg/mL BSA for 1 h at 4 °C, then washed three more times with buffer. In total, 25 µl of washed and blocked beads were added to the sample and incubated for 1 h at 4 °C with continuous mixing to collect the antibody/protein complex. Beads were pelleted by centrifugation for 1 min at 1000 rpm at 4 °C. The bead/antibody/protein complex was then washed with rotation using 1 mL of each buffers twice; low salt immune complex wash buffer (1% SDS, 1% Triton X-100, 2 mM EDTA, 20 mM Tris-HCl pH 8.0, 150 mM NaCl), high salt immune complex wash buffer (1% SDS, 1% Triton X-100, 2 mM EDTA, 20 mM Tris-HCl pH 8.0, 500 mM NaCl), high salt immune complex wash buffer (1% SDS, 1% Triton X-100, 2 mM EDTA, 20 mM Tris-HCl pH 8.0, 500 mM NaCl), TE wash buffer (10 mM Tris-HCl pH 8.0, 1 mM EDTA) and eluted from antibody by adding 250 µl of freshly prepared elution buffer (1% SDS, 0.1 M sodium bicarbonate). We added 5 M NaCl to the elution and cross-linking was reversed by heating at 45 °C overnight followed by the addition of 15 µl of 20 mg/mL RNAase A with 30 min incubation at 37 °C. After this, 10 µl 0.5 M EDTA, 20 µl 1 M Tris-HCl pH 7.5, and 2 µl 20 mg/mL proteinase K were added to the elution and incubated for 2 h at 45 °C. DNA was recovered by phenol/chloroform extraction and ethanol precipitation, using a phenol/chloroform/isoamyl alcohol (25:24:1) mixture twice and chloroform once, then adding 1/10 volume of 3 M sodium acetate pH 5.2, 2 volumes of 100% ethanol, and 1/1000 volume of 20 mg/mL glycogen. Precipitation was allowed to occur overnight at −20 °C. Samples were centrifuged at 13,000 rpm for 30 min at 4 °C, then washed with fresh 80% ethanol, and centrifuged again for 15 min with the same settings. The pellet was air-dried and resuspended in 50 µl nuclease-free water. DNA was purified using Agencourt AMPure XP beads. Libraries were then prepared from this DNA using a KAPA library preparation kit (KK8230) and sequenced on a NovaSeq 6000 machine. FastQC (https://www.bioinformatics.babraham.ac.uk/projects/fastqc/), was used to analyse raw read quality. Any adapter sequences were removed using

Trimmomatic (http://www.usadellab.org/cms/?page=trimmomatic). Bases with Phred quality scores below 25 were trimmed using Sickle (https://github.com/najoshi/sickle). The resulting reads were mapped against the *P. berghei* ANKA genome (v36) using Bowtie2 (version 2.3.4.1). Using Samtools, only properly paired reads with mapping quality 40 or higher were retained and reads marked as PCR duplicates were removed by PicardTools MarkDuplicates (Broad Institute). Genome-wide read counts per nucleotide were normalised by dividing millions of mapped reads for each sample (for all samples, including input) and subtracting input read counts from the ChIP and IgG counts. From these normalised counts, genome browser tracks were generated and viewed using the Integrative Genomic Viewer (IGV).

### Statistical analysis
All statistical analyses were performed using GraphPad Prism 9 (GraphPad Software). Student's *t* test and/or a two-way ANOVA test were used to compare differences between control and experimental groups. Statistical significance is shown as *$*P < 0.05$, $**P < 0.01$, $***P < 0.001$*, ns, not significant. *n* represents the sample in each group or the number of biological replicates. For qRT-PCR, multiple comparisons *t* test, with post hoc test of Holm–Sidak was used to examine significant differences between wild-type and mutant strains.

### Reporting summary
Further information on research design is available in the Nature Portfolio Reporting Summary linked to this article.

## Data availability
The RNA-seq and ChIP-seq data generated in this study have been deposited in the NCBI Sequence Read Archive with accession number: PRJNA808974. Mass spectrometry proteomics data have been deposited to the ProteomeXchange Consortium via the PRIDE partner repository with the dataset identifier PXD043164. Source data are provided with this paper.

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

## Acknowledgements

This work was supported by grants from: ERC advance grant funded by UKRI Frontier Science (EP/X024776/1), MRC UK (G0900109, G0900278, MR/K011782/1) and BBSRC (BB/N017609/1, BB/L013827/1) to R.T. M.Z. was supported by BBSRC (BB/N017609/1) and (EP/X024776/1). The Francis Crick Institute (FC001097), which receives its core funding from Cancer Research UK (FC001097), the UK Medical Research Council (FC001097), and the Wellcome Trust (FC001097) to A.A.H.; the NIH/NIAID (R01 AI136511) and the University of California, Riverside (NIFA-Hatch-225935) to KGLR; E.T. is supported by a personal fellowship from the Nederlandse Organisatie voor Wetenschappelijk Onderzoek, the Netherlands (grant no. VI.Veni.202.223); Swiss National Science

Foundation (31003A_179321 and 310030_208151) to M.B. and R.R. I.M.T. and K.V. acknowledge support by the European Research Council (ERC Synergy Grant, GA Number 855158, granted to IMT), and projects co-financed by the Croatian Government and European Union through the European Regional Development Fund—the Competitiveness and Cohesion Operational Program: IPSted (grant KK.01.1.1.04.0057) and QuantiXLie Center of Excellence (grant KK.01.1.1.01.0004). AE was supported by a Commonwealth Academic Fellowship awarded by the Commonwealth Scholarship Commission in the UK. For Open Access, the authors have applied a CC BY public copyright licence to any Author Accepted Manuscript version arising from this submission. We thank Julie Rodgers for helping to maintain the insectary and other technical works and Cleidiane Zampronio at Warwick University for mass spectrometry methods.

## Author contributions

R.T. conceived and coordinated the project. M.Z., E.R. and R.T. performed and analysed the live-cell imaging, knockdown, and proteomics data; M.Z., R.T. and D.B. performed mice and mosquito-related work; S.A. performed RNA-seq and ChIP-seq analyses; K.V. performed super-resolution STED microscopy; R.M. performed structured illumination microscopy; A.E. and M.Z. performed qRT PCRs; R.R. performed expansion microscopy; A.C.B. performed auxin-inducible degron method of conditional knockdown; A.B. performed mass spectrometry analysis; E.C.T. performed PCA and phylogenetic analysis; D.S.G. performed biostatics analysis; R.T., K.G.L.R., I.M.T., M.B. and A.A.H. coordinated the project; M.Z. and R.T. prepared the first manuscript draft. E.C.T. helped in writing evolutionary part of introduction and results. A.A.H. helped significantly in editing and writing and all authors contributed to manuscript editing and revisions.

## Competing interests

The authors declare no competing interests.

## Additional information

[1]School of Life Sciences, University of Nottingham, Nottingham, UK. [2]Department of Molecular, Cell and Systems Biology, University of California Riverside, 900 University Ave., Riverside, CA, USA. [3]Division of Molecular Biology, Ruđer Bošković Institute, 10000 Zagreb, Croatia. [4]Faculty of Medicine, University of Geneva, Geneva, Switzerland. [5]School of Life Sciences, Gibbet Hill Campus, University of Warwick, Coventry, UK. [6]Department of Genetics and Genome Biology, College of Life Sciences, University of Leicester, Leicester, UK. [7]Malaria Parasitology Laboratory, The Francis Crick Institute, London, UK. [8]Cell Biochemistry, Groningen Biomolecular Sciences and Biotechnology Institute, Faculty of Science and Engineering, University of Groningen, Groningen, The Netherlands. [9]Present address: Department of Medical Biochemistry, Faculty of Basic Medical Sciences, College of Medicine, University of Nigeria, Enugu Campus, Enugu, Nigeria. ✉e-mail: rita.tewari@nottingham.ac.uk

