## [Peer Review file · Nature Communications]

REVIEWER COMMENTS

Reviewer #1 (Remarks to the Author):

The manuscript by Zeeshan et al. investigates the role of aurora-related kinases (ARKs) during plasmodium replication with a focus on ARK2 in the rodent malaria system. The work builds on studies done in the Tewari lab as well as multiple studies in the related Toxoplasma parasites where the functions of TgARK1,2, and 3 have been defined. The function of Ark2 in plasmodium has not been investigated previously, and the current study uses a combination of genetics and microscopy to demonstrate its likely function. The genetics and microscopy are cutting edge. The functional studies in the manuscript are high quality and most of the conclusions are supported by several lines of evidence.

There are some aspects of data presentation that are hard to interpret, especially for a non-expert. These are minor but detract from the exciting findings. The manuscript could be made more readable by including some basic definitions of the components in the introduction.

Overall, this is an interesting study that demonstrates the important role for ARK2 (and EB1) in malaria parasites. This adds new information to our understanding of the process of nuclear division in apicomplexan parasites.

Major comments:

1. Line 245-247. The deconvolution images in S2C do not “confirm” that ARK2 is located on the mitotic spindles. These images “suggest” or “support”, but do not “confirm”. The confirmation requires additional experiments that are done later in the manuscript. This statement could be edited, removed, or just moved further down into the text after the confirmation experiments are done in Fig2 G and H
2. In Figure 1C, the alphafold structure for PfARK2 does not add any information. This is available already at the alphafold website. There is no need to include it here.
3. In Figure 3, the colors in the guide ideally should match the colors in the images below. ARK2 should be green and not red. The color then switches again in B because now Ark2 is labeled with mCherry. These images are complex and difficult to interpret when the colors keep changing. One suggestion would be to define a color scheme and then follow it. The images are all obtained as greyscale and then pseudocolored, so they could be colored however you want. This is true throughout the manuscript.
4. The manuscript is very dense, so any unneeded material that can be removed will improve readability. The failed use of the AID system is not important for the presented findings and can be removed (lines 313-318)

5. Fig 4E the symbols for the two conditions are hard to distinguish. Consider making them bigger or using different symbols.
6. The value of the RNAseq data is unclear. Lines 357-380 and figure H and I do not add that much information to the functional information. Removal or shortening or movement to supplemental should be considered.
7. In Fig 5B, the definitions for the data are not provided. What criteria were used for determining that a protein was background? What criteria were used for significance?
8. Line 453-461 – the localization of EB1-GFP in zygote to ookinete should be shown. However, this paragraph includes several statements that are only loosely backed up by the single series of images. If the accumulation at the nascent apical end is real, this should be quantified; in what percentage of zygotes to ookinete (or retorts) is this present? Similarly, the statement about association with sub-pellicular MTs is not supported by the data. These should be removed or more carefully supported by additional data.
9. Fig 7F is a different format from 4H. The data presentation should be harmonized in the manuscript.
10. Lines 502-507 – the text is very informal in this section. “Subtle but clear co-variation” is not defined. For a reader that is not an expert in analysis of this data, the PCA does not show how meaningful the associations are or if they are significant or reproducible.
11. Figure 8D is very good and helps to build an understandable model. This should be harmonized with the guides in earlier figures. For example, the colors should match. Also, in figure 2 the intranuclear MTOC is referred to as the acentriolar MTOC. In Figure 8D, it is referred to as intranuclear MTOC. This is confusing.
12. A general question: if Ark2 and EB1 are binding to microtubules, why did the tubulins not come up in the immunoprecipitation experiments? Was this an expected or unexpected finding? This should be discussed in the manuscript.

Minor comments:

1. Line 51: the grammar in the abstract should be improved for easier readability. The sentence, while technically correct, seems to be missing a few words.
2. In Table S1: the common gene names should be used, or at least included. For example, PBANKA_123456 or TGME49_123456). The same holds for the other organisms.
3. Line 193 – ApiARK needs to be defined.
4. Line 835 – what is Vectashield 19? This must be a typographical error.
5. In the guide for figure 2: the color of the acentriolar MTOC and the spindle are too similar to distinguish in the diagram.
6. The videos S1 through S3 are very short. It would be better to have a longer video that shows the changes throughout the process. Same thing is true for S4-S5; S6-S7.

Reviewer #2 (Remarks to the Author):

This manuscript by Zeeshan et al investigates the role of the atypical Aurora paralog, ARK2, during male gametogeny in the rodent malaria model, *Plasmodium berghei*.

The authors have generated ARK2-GFP parasites and found that tagging did not affect growth. They report that ARK2 is associated with the spindle during male gametogenesis; and report the interesting finding that division is asynchronous.

The authors provide images of the location of ARK2-mCherry with respect to NDC80-GFP.

Line 249. The authors state: "The location of both ARK2-mCherry and NDC80-GFP was next to the stained DNA, and with a partial overlap, although NDC80-GFP was always closer to the DNA (Fig 2C and Fig S3A)." This statement doesn't capture the observation that the ARK2-mCherry signal extends beyond the NDC80-GFP signal. Similarly Figure 2A, 3rd panel illustrates the kinetochore as being distributed along the full length of the ARK2-mCherry-decorated spindle. This is not supported by the data. It would be useful to provide a quantitative analysis of the overlap at different stages. It may be useful to nuance the conclusion of a functional (direct) interaction of ARK2-mCherry and NDC80-GFP given their different distributions.

Similarly, in the analysis of the location of ARK2-GFP during the zygote to ookinete transition there is little overlap of the ARK2-GFP and NDC80-GFP signals. The authors should comment on the implication of this for the role of ARK2 in the meiotic spindle.

Interestingly ARK2 down-regulation had no effect on mitosis in male gamete formation (exflagellation) nor meiosis in zygote differentiation (ookinete development). However, it was associated with up-regulation of a number of motor proteins, suggesting a partial defect that may be compensated by increased expression of other proteins, such that it does not manifest phenotypically until later in the cell cycle.

Line 372. Note: Fig 4B should presumably be Fig 4H.

Protein interaction studies provided some evidence for the interaction of the ARK2 with MISFIT and EB1. The authors provide CHIP-Seq and live cell imaging of ARK2/EB1 co-transfectants that confirms a high level of overlap of location. Again, it would be useful to quantitate the level of co-localisation in the fluorescence microscopy data.

Line 430. The authors state that “.. parasite lines expressing EB1-mCherry and the basal body marker SAS4-GFP (Zeeshan et al., 2022) showed EB1’s association with the formation of basal bodies that serves as the

MT organising centre for axonemes (Fig 6E).” It is not clear what is the basis for this statement. There is no evident overlap of EB1 and SAS4 signals. They are presumably on opposite sides of the nuclear membrane.

The authors report the very interesting finding that EB1 is not required from asexual mitosis, male gamete formation or ookinete development; but resulted in a decreased number of oocysts.

Immunoprecipitation of EB1 revealed a large network of interactions, including potential interactions with myosin K, MISFIT and ARK2 in activated male gametocytes. Given that EB1 is very likely bound to microtubules, the immunoprecipitation protocol is likely to bring down microtubule fragments and a number of associated proteins. Thus, immunoprecipitation does not provide sufficient evidence to assume a direct interaction.

Therefore, while the experiments are extensive and carefully performed, I am not convinced that the data provide sufficiently strong support for the author’s conclusion that “ARK2 and EB1 are part of a regulatory axis that likely includes MyoK and MISFIT located at, or near the spindle MT-kinetochore interface and are likely involved in the rapid cycles of spindle assembly and chromosome segregation during male gametogony.”

At this stage, the data support an association of ARK2 and EB1, but it is not yet clear if it is a direct or indirect association. And while ARK2 and EB1 are individually required for male gametogony, the proposed ARK2-EB1 axis remains a good hypothesis rather than an established structure. I suggest that the authors revisit the title and conclusions to make it clearer that additional work is required.

Nonetheless, the work provides interesting insights into the process of male gametogenesis in Plasmodium.

Minor points:

Line 183. The authors state: "Similarly, PfARK1 is located at, or near the spindle pole (Reininger et al., 2011) suggesting that apicomplexan ARK1 is the centromere-based equatorial-like AK." This seems contradictory. The contradiction is due to the fact that the Reininger et al study did not have sufficient resolution to distinguish centromeres from spindle poles. The sentence could be rephrased.

Figure 2A, 3rd panel. It would be useful to make it clearer that the spindle is inside the nucleus.

Reviewer #3 (Remarks to the Author):

Overview:

In this manuscript Zeeshan et al have set out to deepen our understanding of mechanisms underlying mitotic and meiotic cell division in Plasmodium parasites. Specifically, the authors focus on the spindle-associated aurora kinases in Plasmodium berghei during sexual development inside the Anopheles mosquito. The authors use a suitable combination of methods, including; live-cell fluorescence imaging, mass spec, super-resolution microscopy, as well as ChIP-seq, to characterize ARK2 (an atypical Aurora kinase paralog), its localization during gametogony and meiosis including speculation regarding its function during these events. They further provide data that indicates that ARK2 and EB1 have complementary functions during endomitotic division in Plasmodium.

The study addresses a fundamental topic in biology and here, specifically in Plasmodium biology, linked to broadening our understanding of the mechanisms involved in cell division. Further it adds important information of how ARK2 is involved in the rapid process of gametogony and also meiosis inside the mosquito vector.

It seems, however, that the manuscript could use a thorough revision in order to highlight the most important findings (the current version includes 8 figures, where some of these might be better suited as supplemental figures). Several explanations/clarifications appear to be missing throughout the manuscript text, which makes it a bit difficult to follow the rationale of methods or analysis tools used, as well as the descriptions of certain parts of the results section. Some of which are outlined below.

Based on the background information in the manuscript, it is not clear why the authors did not also assay the potential involvement of PbARK1 and PbARK3 during gametogony and meiosis. It would be good if the authors could account for this.

The authors describe that “all three Plasmodium ARKs are likely essential for proliferation in asexual blood stage schizogony”, but there is no rationale for not including PbARK1 and PbARK3 in this study. That would elevate the significance of the study by describing how the three kinases are coordinated in *P. berghei* during cell division in the midgut of the mosquito.

Comments:

-Figures; 1, 5 and potentially 7 may be better fits as supplementary figures

-Corrections of language in the abstract and throughout the manuscript is necessary

i.e. line 31 “suggesting THAT the”

line 39 “THE mosquito host”

line 41 “Kinome studies HAVE”

line 47 “using THE rodent”

Line 56: remove “Plasmodium”

etc..

The first section of the results relating to the evolutionary/phylogenetic analyses are quite unclear in its description including the specific references to Figure 1. The conclusion that the title suggests is not evident in the text or in Figure 1, beyond the size of PbARK2 and 3, which appear to be in the range of that of *Toxoplasma*. Further, the title should be more specific than “evolutionary history” and the choice of colors needs to be revised since it is impossible to read some of the text and to make out the difference between e.g. 3 and 4 in Fig 1A, relating to the spindle pole body and central spindle, respectively.

-The statement that the authors “constructed new and previously established phylogenetic profiles” doesn’t make any sense the way it is written.

-Line 174 to 176: it's not clear how this mapping was performed. The authors refer to a review by Hochegger 2013, however, a description of identifying these conserved patterns of AKs sub-functionalization to eukaryotes is required, as to how different AKs are mapped to the five mentioned sub-cellular locations. This section needs more clarification or to be included in the methods section.

It appears that there should be a description why *P. falciparum* is used for the AlphaFold structural prediction of ARK2, since it is not included in the phylogenetic diagram.

Figure 1 and the section relating to it in the manuscript need to be fully revised.

-Line 119-123: Note that previous gene expression studies indicate that ARK2 appears to be involved in proliferation and cell division in both Pf and Pb, PMID: 30409996 and previously reviewed in PMID: 23462523, although these studies did not describe localization. However, this should be added to the description in the introduction.

-Line 120: makes reference to Berry et al, 2016, to claim that ARK1 and ARK3 was limited to asexual blood stages in *P. falciparum*. However, the authors describe that they determine the localization of the TgArk3 orthologue in *P. falciparum*, it does not specifically make a claim about ARK1. However, it is stated that PfARK2 is restricted to Plasmodium species.

-Line 124-126: the statement about schizogony requires a reference. If it is the case that the mitotic process differs substantially between the asexual stage and male gametogenesis the authors will need to clarify this further. Is it only the timing that differs or are there differences in the process?

-Line 130: to clarify add number of resulting microgametes. Also, a segue into the following statement about meiosis seems appropriate, and Meiosis occurs well within 24h, clarify this to the broader community.

-Line 133: to put things into perspective for the broader community, it will likely be good to add a ballpark estimation of the number of sporozoites produced by the oocysts

-Line 138 "*Plasmodium*" (italic)

-Line 147: replace "associated with a novel protein complex.." with "associated with a previously undescribed protein complex.."

-Lines 328-333: The authors provide the percentage of ookinete conversion in Figure 4B, which appears similar to that of the WT, however it would be informative if the authors also provided imaging data indicating whether the Pclag-ARK2 appear abnormal/deformed in their morphology.

-Lines 370-372: the authors mention differentially expressed genes in Pclag-ARK2 gametocytes and refer to figure 4B and Table S2. Figure 4B shows % ookinete conversion. The authors are likely referring to Figure 4H?

-Lines 390 to 395: It is not well described how the PCA was performed, for the broad reader it seems necessary to further describe the rationale behind performing this analysis and more background as to how the comparison is done.

-Line 479: (Associated with figure 7E) The authors visualized only custom tracks, but do not provide count reads. Since there is a possibility that EB1 transcripts are still produced by the EB1-KO parasites it would be good to also provide a direct link for custom tracks in order for the reader to investigate the reads coverage thoroughly.

Further, the authors did not provide an indicator for the adj P-value in Figure 7F

Line 481: It would be good to emphasize how the GO enrichment analysis was performed in this section (Note Fig S10 is absent).

Line 1329: Fig S10 cannot be found in the supplementary materials

Reviewer's Comments:

Reviewer #1 (Remarks to the Author)

The manuscript by Zeeshan et al. investigates the role of aurora-related kinases (ARKs) during plasmodium replication with a focus on ARK2 in the rodent malaria system. The work builds on studies done in the Tewari lab as well as multiple studies in the related Toxoplasma parasites where the functions of TgARK1,2, and 3 have been defined. The function of Ark2 in plasmodium has not been investigated previously, and the current study uses a combination of genetics and microscopy to demonstrate its likely function. The genetics and microscopy are cutting edge. The functional studies in the manuscript are high quality and most of the conclusions are supported by several lines of evidence.

There are some aspects of data presentation that are hard to interpret, especially for a non-expert. These are minor but detract from the exciting findings. The manuscript could be made more readable by including some basic definitions of the components in the introduction.

Overall, this is an interesting study that demonstrates the important role for ARK2 (and EB1) in malaria parasites. This adds new information to our understanding of the process of nuclear division in apicomplexan parasites.

Response: we thank the reviewer for their constructive feedback, which we have used to improve the presentation of the data. We address the concerns below.

Major comments:

1. Line 245-247. The deconvolution images in S2C do not “confirm” that ARK2 is located on the mitotic spindles. These images “suggest” or “support”, but do not “confirm”. The confirmation requires additional experiments that are done later in the manuscript. This statement could be edited, removed, or just moved further down into the text after the confirmation experiments are done in Fig2 G and H

Response: the sentence is now modified: ‘these images suggest’. Please see line number 279.

2. In Figure 1C, the alphafold structure for PfARK2 does not add any information. This is available already at the alphafold website. There is no need to include it here.

Response: We appreciate the reviewer's comment. We originally included the predicted structure of ARK2 to underline the extensive sequence divergence and expansion (in terms of length) of this Aurora kinase paralogue, and to indicate that there are no additional structural domains predicted. Considering the simplification of the manuscript proposed by the other reviewers, we moved Figure 1C to the

supplementary material, and now it is in Supplementary Figure S1. In addition, we have used ColabFold (online AlphaFold2) to predict the structure of *P. berghei* ARK2.

3. In Figure 3, the colors in the guide ideally should match the colors in the images below. ARK2 should be green and not red. The color then switches again in B because now Ark2 is labeled with mCherry. These images are complex and difficult to interpret when the colors keep changing. One suggestion would be to define a color scheme and then follow it. The images are all obtained as greyscale and then pseudocolored, so they could be colored however you want. This is true throughout the manuscript.

Response: We apologise for the confusion and have changed the colours of the guide to match those of the images. We have generated parasite lines expressing ARK2-GFP (green) and ARK2-mCherry (red) (by genetic modification of the endogenous locus) and crossed them with lines expressing different tagged cell division markers. In Fig. 3A, we use an ARK2-GFP (green) line to study the location of ARK2 during ookinete development. In Fig 3B, we use an ARK2-mCherry (red) line crossed with one expressing kinetochore marker, NDC80-GFP (green) to examine colocalization. Please see the modified Fig. 3. We provide a key above the respective figures.

4. The manuscript is very dense, so any unneeded material that can be removed will improve readability. The failed use of the AID system is not important for the presented findings and can be removed (lines 313-318)

Response: We have shortened the Introduction and integrated Figure 1A/B into this part of the manuscript. We have moved several elements of the manuscript to the supplementary material to improve readability: we have moved the evolutionary analyses (former Figure 1) to the supplementary material as Note 1, together with the introduction and the text part of ARK2 localization at various life cycle stages, as Note 2. With all due respect to the reviewer's opinion, we would like to keep the short section on the failed use of the AID system, because it indicates that any conditional knockdown approach may not be successful and the use of two conditional systems gives greater significance to the study.

5. Fig 4E the symbols for the two conditions are hard to distinguish. Consider making them bigger or using different symbols.

Response: This has been improved; please see modified Fig. 4.

6. The value of the RNAseq data is unclear. Lines 357-380 and figure H and I do not add that much information to the functional information. Removal or shortening or movement to supplemental should be considered.

Response: We analysed the transcriptome of the *P_{clag}ark2* parasite because the *ark2* downregulation had no visible effect on either mitosis in male gamete formation

(exflagellation) or meiosis in zygote differentiation (ookinete development), although there was a phenotype at the oocyst stage due to a defect conferred by the male gamete allele as shown in Fig. 4G. RNAseq analysis indicated an associated up-regulation of other genes including aurora kinases (ark1 and ark3), and a number of motor and kinetochore proteins, suggesting a partial defect that may be compensated by increased expression of other proteins, and therefore not manifest phenotypically until later in the life cycle. This hypothesis is supported by evidence in Fig. 4H and I. Other RNAseq data are shown in supplementary Fig. 7. We think it is important to keep Figs 4H and 4I in a main figure, and hope that the reviewer agrees with this.

7. In Fig 5B, the definitions for the data are not provided. What criteria were used for determining that a protein was background? What criteria were used for significance?

Response: We apologize to the reviewer for the confusing use of the term 'background'. In our figures that include PCA plots, the datapoints referred to as 'background' included common proteins that are often found in the background of proteins in pulldown experiments. However, in this instance 'background' referred to any non-annotated datapoints in the figure. We agree this is confusing and therefore we have removed this term altogether. We changed the legend of the figures to make clearer the selection of specific protein groups. We have also removed the annotation 'background' in supplementary Table 3. In addition, we have added a subsection to the methods to explain in more detail how PCA analysis of the pulldown experiments was performed. A part of the methods section that had been inadvertently deleted before manuscript submission, has also been restored.

PCA is a data exploration tool, and no p-value statistic can be calculated to assess the statistical significance of the validity of each principal component in the analysis. Therefore, we cannot provide a significance value as such. In addition, we performed only two pulldown experiments to minimize animal use. To indicate more clearly how we used the PCA (as an exploration tool) we have modified Fig. 5, adding a short table to indicate and clarify the peptide values upon which the PCA was performed. We have modified the text in the results section to soften our claims surrounding a putative interaction between EB1 and ARK2 and present a more nuanced view on this. Please see modified figures 5 and 8 and the text on the EB1-GFP pull down results at line numbers 598-618.

8. Line 453-461 – the localization of EB1-GFP in zygote to ookinete should be shown. However, this paragraph includes several statements that are only loosely backed up by the single series of images. If the accumulation at the nascent apical end is real, this should be quantified; in what percentage of zygotes to ookinete (or retorts) is this present? Similarly, the statement about association with sub-pellicular MTs in not supported by the data. These should be removed or more carefully supported by additional data.

Response: The location of EB1-GFP during the zygote to ookinete transition is shown in supplementary Fig. 10A. We have included the data to quantify the apical end localization and added these as supplementary Fig 10B. We have removed the statement about an association with sub-pellicular MTs. Please see the line numbers 537-545.

9. Fig 7F is a different format from 4H. The data presentation should be harmonized in the manuscript.

Response: Thanks for the suggestion. The presentation of data in Fig. 4H has now been changed to a volcano plot to match Fig. 7F.

10. Lines 502-507 – the text is very informal in this section. “Subtle but clear co-variation” is not defined. For a reader that is not an expert in analysis of this data, the PCA does not show how meaningful the associations are or if they are significant or reproducible.

Response: We agree with the reviewer that PCA is not an indication of significance per se, but of reproducibility, and usually other tools are needed to validate interpretation of the clusters that result from PC analyses. We have removed the “subtle but clear co-variation” statement. Please also see the response to point 7 above, on the use of PCA and statistical significance.

11. Figure 8D is very good and helps to build an understandable model. This should be harmonized with the guides in earlier figures. For example, the colors should match. Also, in figure 2 the intranuclear MTOC is referred to as the acentriolar MTOC. In Figure 8D, it is referred to as intranuclear MTOC. This is confusing.

Response: We apologise for the confusion. We have now harmonised the guide to Fig. 2 with that of Fig. 8. We now use acentriolar MTOC throughout the manuscript/figures and have removed the ‘intranuclear’ term to avoid any confusion.

12. A general question: if Ark2 and EB1 are binding to microtubules, why did the tubulins not come up in the immunoprecipitation experiments? Was this an expected or unexpected finding? This should be discussed in the manuscript.

Response: We thank the reviewer for this comment. Tubulins are always present in all our pulldown experiments. However, using PCA the alpha and beta tubulin subunit peptide values show a clear co-variation (they cluster) over all experiments (EB1/ARK2/GFP pulldown) indicating that they are likely functionally and/or physically associated. There are some small differences in tubulin pulldown per experiments, but these are not to the extent that we can make any relevant conclusions. We have now indicated the position of tubulin in Fig. 8A.

Minor comments:

1. Line 51: the grammar in the abstract should be improved for easier readability. The sentence, while technically correct, seems to be missing a few words.

Response: We have checked this text for readability and hope that it now clear.

2. In Table S1: the common gene names should be used, or at least included. For example, PBANKA_123456 or TGME49_123456). The same holds for the other organisms.

Response: we have now added all the commonly used gene names we could find for proteins in the table and have updated identifiers to avoid any confusion over which genes are annotated.

3. Line 193 – ApiARK needs to be defined.

Response: we apologize to the reviewer for using the poorly defined term ApiARK, and have now replaced it with apicomplexan aurora paralogue ARK2 etc. This section of text is now moved to the supplementary material.

Line 835 – what is Vectashield 19? This must be a typographical error.

Response: Thanks for pointing it out. We have now corrected it to 'Vectashield'.

5. In the guide for figure 2: the color of the acentriolar MTOC and the spindle are too similar to distinguish in the diagram.

Response: We have modified the colours to differentiate the structures more clearly. Please see the modified Fig. 2.

6. The videos S1 through S3 are very short. It would be better to have a longer video that shows the changes throughout the process. Same thing is true for S4-S5; S6-S7.

Response: We have tried to produce longer videos, but GFP/RFP gets photobleached after 2 to 3 min. We were unable to make time lapses videos for the whole process that takes 10 to 12 min. We have focused largely on the organisation of the spindle for the first round of mitosis, a process which is replicated at 4 to 5 min and then at 7 to 8 min.

Reviewer #2 (Remarks to the Author)

This manuscript by Zeeshan et al investigates the role of the atypical Aurora paralog, ARK2, during male gametogeny in the rodent malaria model, *Plasmodium berghei*.

The authors have generated ARK2-GFP parasites and found that tagging did not affect growth. They report that ARK2 is associated with the spindle during male gametogenesis; and report the interesting finding that division is asynchronous.

The authors provide images of the location of ARK2-mCherry with respect to NDC80-GFP.

Line 249. The authors state: "The location of both ARK2-mCherry and NDC80-GFP was next to the stained DNA, and with a partial overlap, although NDC80-GFP was always closer to the DNA (Fig 2C and Fig S3A)." This statement doesn't capture the observation that the ARK2-mCherry signal extends beyond the NDC80-GFP signal. Similarly, Fig. 2A, 3rd panel illustrates the kinetochore as being distributed along the full length of the ARK2-mCherry-decorated spindle. This is not supported by the data. It would be useful to provide a quantitative analysis of the overlap at different stages. It may be useful to nuance the conclusion of a functional (direct) interaction of ARK2-mCherry and NDC80-GFP given their different distributions.

Response: We agree with the reviewer and recognise the limitation of widefield microscopy. That is why we tried to extend this observation using high resolution STED, SIM and Expansion microscopy. Analysis of live cell images and time lapse imaging clearly showed the relative orientation of ARK2, NDC80 and nuclear DNA. STED and expansion microscopy clearly showed the distribution of ARK2 along the full length of the spindle (Fig. 2G, H). SIM images showed the beaded NDC80 decoration with the ARK2 signal extending beyond the NDC80 signal (Fig. 2I). Now we have added some data quantifying the amount of overlap between ARK2 and NDC80 at different stages of gametogony and show the variable Pearson's colocalization coefficient. Please see the modified Fig. 2. We have also modified the text in the conclusions section to describe a putative association rather than a direct interaction of ARK2 and NDC80. Please see the conclusion in line numbers 791-798.

Similarly, in the analysis of the location of ARK2-GFP during the zygote to ookinete transition there is little overlap of the ARK2-GFP and NDC80-GFP signals. The authors should comment on the implication of this for the role of ARK2 in the meiotic spindle.

Response: We have quantified the overlap between ARK2 and NDC80 signals during the different stages of ookinete development. Please see modified Fig. 3B. The dynamic locations of ARK2 (spindle) and NDC80 (kinetochore) demonstrate chromosome segregation occurring in meiosis during ookinete differentiation. We have added text on the implication of their locations in line numbers 503-508.

Line 372. Note: Fig 4B should presumably be Fig 4H.

Response: We thank the reviewer for spotting this error, which has now been corrected.

Protein interaction studies provided some evidence for the interaction of the ARK2 with MISFIT and EB1. The authors provide CHIP-Seq and live cell imaging of ARK2/EB1 co-transfectants that confirms a high level of overlap of location. Again, it would be useful to quantitate the level of co-localisation in the fluorescence microscopy data.

Response: We have quantified the overlap between ARK2 and EB1 signals. Please see the modified Fig. 6D.

Line 430. The authors state that “.. parasite lines expressing EB1-mCherry and the basal body marker SAS4-GFP (Zeeshan et al., 2022) showed EB1’s association with the formation of basal bodies that serves as the MT organising centre for axonemes (Fig 6E).” It is not clear what is the basis for this statement. There is no evident overlap of EB1 and SAS4 signals. They are presumably on opposite sides of the nuclear membrane.

Response: The sentence has been modified to ‘...the basal body (SAS4) in cytoplasm and spindle dynamics (EB1) in the nucleus are co-ordinated in time and space across the two compartments during male gametogenesis.

The authors report the very interesting finding that EB1 is not required from asexual mitosis, male gamete formation or ookinete development; but resulted in a decreased number of oocysts.

Immunoprecipitation of EB1 revealed a large network of interactions, including potential interactions with myosin K, MISFIT and ARK2 in activated male gametocytes. Given that EB1 is very likely bound to microtubules, the immunoprecipitation protocol is likely to bring down microtubule fragments and a number of associated proteins. Thus, immunoprecipitation does not provide sufficient evidence to assume a direct interaction.

Therefore, while the experiments are extensive and carefully performed, I am not convinced that the data provide sufficiently strong support for the author’s conclusion that “ARK2 and EB1 are part of a regulatory axis that likely includes MyoK and MISFIT located at, or near the spindle MT-kinetochore interface and are likely involved in the rapid cycles of spindle assembly and chromosome segregation during male gametogony.”

At this stage, the data support an association of ARK2 and EB1, but it is not yet clear if it is a direct or indirect association. And while ARK2 and EB1 are individually required for male gametogony, the proposed ARK2-EB1 axis remains a good

hypothesis rather than an established structure. I suggest that the authors revisit the title and conclusions to make it clearer that additional work is required.

Response: We thank the reviewer for this constructive comment. We agree that our data provides circumstantial but not direct evidence for a physical interaction between EB1 and ARK2. We have therefore removed any mention of a direct interaction between ARK2 and EB1 (or MyoK and MISFIT) and made it clear that this remains a hypothesis. In a recent study by Yang et al., 2023 similar findings describe the association of EB1 with the spindle and lateral attachment with the kinetochore. In view of the reviewer's concern, we make this hypothesis more explicit in the discussion and have modified the title and conclusion of the manuscript.

Nonetheless, the work provides interesting insights into the process of male gametogenesis in Plasmodium.

We thank the reviewer for these thoughtful comments.

Minor points:

Line 183. The authors state: "Similarly, PfARK1 is located at, or near the spindle pole (Reininger et al., 2011) suggesting that apicomplexan ARK1 is the centromere-based equatorial-like AK." This seems contradictory. The contradiction is due to the fact that the Reininger et al study did not have sufficient resolution to distinguish centromeres from spindle poles. The sentence could be rephrased.

Response: We agree with the reviewer. Now the sentence is rephrased as "PfARK1 is located at, or near the spindle pole (Reininger et al., 2011). This text has now been moved to the supplementary material.

Figure 2A, 3rd panel. It would be useful to make it clearer that the spindle is inside the nucleus.

Response: We have tried to improve this figure based on the reviewer's comment. Please see modified Fig. 2A.

Reviewer #3 (Remarks to the Author):

Overview:

In this manuscript Zeeshan et al have set out to deepen our understanding of mechanisms underlying mitotic and meiotic cell division in *Plasmodium* parasites. Specifically, the authors focus on the spindle-associated aurora kinases in *Plasmodium berghei* during sexual development inside the *Anopheles* mosquito. The authors use a suitable combination of methods, including; live-cell fluorescence imaging, mass spec, super-resolution microscopy, as well as ChIP-seq, to characterize ARK2 (an atypical Aurora kinase paralog), its localization during gametogony and meiosis including speculation regarding its function during these events. They further provide data that indicates that ARK2 and EB1 have complementary functions during endomitotic division in *Plasmodium*. The study addresses a fundamental topic in biology and here, specifically in *Plasmodium* biology, linked to broadening our understanding of the mechanisms involved in cell division. Further it adds important information of how ARK2 is involved in the rapid process of gametogony and also meiosis inside the mosquito vector.

It seems, however, that the manuscript could use a thorough revision in order to highlight the most important findings (the current version includes 8 figures, where some of these might be better suited as supplemental figures). Several explanations/clarifications appear to be missing throughout the manuscript text, which makes it a bit difficult to follow the rationale of methods or analysis tools used, as well as the descriptions of certain parts of the results section. Some of which are outlined below.

Based on the background information in the manuscript, it is not clear why the authors did not also assay the potential involvement of PbARK1 and PbARK3 during gametogony and meiosis. It would be good if the authors could account for this.

Response: We appreciate the reviewers' concern about not including studies on PbARK1 and PbARK3. They were not included for two reasons. First, we know that all three PbARKs are essential for *P. berghei* asexual blood stage development. PbARK1 (Reininger et al., 2011) and PbARK3 (Berry et al., 2016) have been located next to the DNA at discrete foci in mature blood stage parasites, but PbARK2 was unexplored. In the current study we found a similar location of ARK2 in blood stages next to the DNA within the nucleus at discrete foci and that it shows an interesting dynamic location during gametogony. Secondly, the pull-down data revealed an atypical association of PbARK2 with microtubule and kinetochore proteins in gametocytes, which motivated us to study ARK2 and microtubule binding protein EB1 in detail and to explore this unusual association. We have carried out separate studies on ARK1 and ARK3 but these also need further in-depth analysis. We explain in the introduction why we focused only on PbARK2; please see the text in line numbers 130-138.

We hope that the reviewer will appreciate that this detailed study, including the in-depth analysis of the conditional knockdown of ARK2 and its scaffold proteins, is already a huge amount of work and it would be unrealistic to include similar studies of all three ARKs in a single manuscript. We are planning to perform such in-depth studies on ARK1 and ARK3 but these are not within the scope of this manuscript.

The authors describe that “all three Plasmodium ARKs are likely essential for proliferation in asexual blood stage schizogony”, but there is no rationale for not including PbARK1 and PbARK3 in this study. That would elevate the significance of the study by describing how the three kinases are coordinated in *P. berghei* during cell division in the midgut of the mosquito.

Response: Please see the comments above on PbARK1 and PbARK3. We believe that these proteins need more in-depth analysis but that this is outside the scope of this study. It would of course be of interest in the future, as the reviewer suggests, to describe any coordination between these three kinases, but at the moment there is little evidence of such coordination.

Comments:

-Figures; 1, 5 and potentially 7 may be better fits as supplementary figures

Response: As suggested by the reviewer, parts of these figures have been moved to supplementary material, while retaining the major focus in the main figures. Please see the modified Figs. 1 and 5.

-Corrections of language in the abstract and throughout the manuscript is necessary
i.e. line 31 “suggesting THAT the”
line 39 “THE mosquito host”
line 41 “Kinome studies HAVE”
line 47 “using THE rodent”
Line 56: remove “Plasmodium”
etc..

Response: The language has been checked throughout the manuscript and corrected where necessary.

The first section of the results relating to the evolutionary/phylogenetic analyses are quite unclear in its description including the specific references to Figure 1. The conclusion that the title suggests is not evident in the text or in Figure 1, beyond the size of PbARK2 and 3, which appear to be in the range of that of *Toxoplasma*.

Response: we regret that the reviewer did not appreciate our evolutionary analysis. To shorten the manuscript, we have clarified and moved part of this text to

supplementary material. Some of this material also formed part of an introduction to the study so we integrated those elements into the introductory text.

However, with respect, we do disagree with the reviewer on the issue of divergence – for the following reasons:

(1) the parallel evolution of Aurora kinases suggests that the equatorial kinase function (as part of the chromosomal passenger complex [CPC]) remains constant, while paralogues that arise at specific nodes in the tree of life appear to specialize either towards the spindle (for instance, AURKA in humans) or other functions (for example AURKC is meiosis-specific and there are different functions for Aurora paralogues in trypanosomes). In this light it might be expected that both ARK2 and ARK3 have a functional role at the spindle apparatus, or an entirely different role altogether.

(2) For *Toxoplasma gondii*, an apicomplexan parasite that shares all the Aurora duplications with Plasmodium (so the three Aurora kinases were present in their common ancestor), its ARK2 paralogue is present at the spindle, making it very likely that Plasmodium ARK2 would also be at the spindle. A functional commonality for ARK2 between *Toxoplasma gondii* and *Plasmodium berghei* was to be expected.

(3) There are no clear Aurora kinase (spindle) scaffolds present other than INCENP1/2. This, together with the extended length of both ARK2 and ARK3, suggests that Aurora kinase signalling in Apicomplexa is highly divergent.

Further, the title should be more specific than “evolutionary history”

Response: We have changed the title of this section to: “Comparative genomics in combination with subcellular localisation of Aurora kinase paralogues throughout the eukaryotic tree of life indicate that Plasmodium ARK2 is a putative spindle-based Aurora kinase”. Please see the text in the supplementary material.

and the choice of colors needs to be revised since it is impossible to read some of the text and to make out the difference between e.g. 3 and 4 in Fig 1A, relating to the spindle pole body and central spindle, respectively.

Response: we have now replaced the colour for the central spindle with a blue tint to avoid confusion. In addition, we apologize for submitting what was probably a low resolution version of the figure that lacked clarity.

-The statement that the authors “constructed new and previously established phylogenetic profiles” doesn’t make any sense the way it is written.

Response: We apologize for any confusion arising from our use of language. We intended to state that we constructed phyletic profiles of the proteins present in Fig 1B based on previously established phylogenetic analyses, as well as our new analyses. We have now changed this sentence to: “We therefore updated previously

established phyletic profiles (Komaki et al., 2022; Kops et al., 2020; van Hooff et al., 2017) for AKs, related mitotic kinases and their location-specific scaffolds and activators in a selection of eukaryotes (Fig. 1B, Supplementary Table 1).” This section of text is now present in the supplementary material.

-Line 174 to 176: it's not clear how this mapping was performed. The authors refer to a review by Hochegger 2013, however, a description of identifying these conserved patterns of AKs sub-functionalization to eukaryotes is required, as to how different AKs are mapped to the five mentioned sub-cellular locations. This section needs more clarification or to be included in the methods section.

Response:

We apologise for being unclear here. This mapping was based purely on a careful search of the literature, which mostly overlaps with the Hochegger 2013 review. All the papers we uncovered are referenced in either the introduction or the revised supplementary notes on the the evolutionary history of the Aurora kinase family.

It appears that there should be a description why *P. falciparum* is used for the AlphaFold structural prediction of ARK2, since it is not included in the phylogenetic diagram.

Response: We have moved Fig.1C to the supplementary material. We have used ColabFold (AlphaFold2) to predict the structure of *PbARK2*.

Figure 1 and the section relating to it in the manuscript need to be fully revised.

Response: We have modified extensively Fig. 1 and related text, moving most of the modifications to the supplementary material and some to the introduction. Please see modified Fig. 1 and the supplementary text.

-Line 119-123: Note that previous gene expression studies indicate that ARK2 appears to be involved in proliferation and cell division in both Pf and Pb, PMID: 30409996 and previously reviewed in PMID: 23462523, although these studies did not describe localization. However, this should be added to the description in the introduction.

Response: We have added both references in the revised manuscript. Please see the introduction line number 136.

-Line 120: makes reference to Berry et al, 2016, to claim that ARK1 and ARK3 was limited to asexual blood stages in *P. falciparum*. However, the authors describe that

they determine the localization of the TgArk3 orthologue in *P. falciparum*, it does not specifically make a claim about ARK1. However, it is stated that PfARK2 is restricted to Plasmodium species.

Response: We have modified the sentences describing the localization studies of PfARK1 by Reininger et al., 2011 and PfARK3 by Berry et al 2006. Please see the introduction lines 131-134.

-Line 124-126: the statement about schizogony requires a reference. If it is the case that the mitotic process differs substantially between the asexual stage and male gametogenesis the authors will need to clarify this further. Is it only the timing that differs or are there differences in the process?

Response: We have provided the references for schizogony and other stages. We have described the differences in the mitotic process between the asexual stage and gametogenesis. Please see them in line number 143-145.

-Line 130: to clarify add number of resulting microgametes. Also, a segue into the following statement about meiosis seems appropriate, and Meiosis occurs well within 24h, clarify this to the broader community.

Response: We have added there are eight microgametes at line number 148. We have now modified the statements and clarified the two events for the broader community. Please see the line numbers 148-156.

-Line 133: to put things into perspective for the broader community, it will likely be good to add a ball-park estimation of the number of sporozoites produced by the oocysts

Response: We have added the number (in thousands) of sporozoites produced in each oocyst. Please see the line number 156.

-Line 138 "*Plasmodium*" (italic)

Response: This text has been revised.

-Line 147: replace "associated with a novel protein complex.." with "associated with a previously undescribed protein complex.."

Response: Text revised as suggested.

-Lines 328-333: The authors provide the percentage of ookinete conversion in Figure 4B, which appears similar to that of the WT, however it would be informative if the

authors also provided imaging data indicating whether the Pclag-ARK2 appear abnormal/deformed in their morphology.

Response: We now provide the images so that the reader can assess the morphology. Please see modified Fig. 4B.

-Lines 370-372: the authors mention differentially expressed genes in Pclag-ARK2 gametocytes and refer to figure 4B and supplementary Table 2. Figure 4B shows % ookinete conversion. The authors are likely referring to Fig.4H?

Response: Thanks for pointing out this error, which is now corrected.

-Lines 390 to 395: It is not well described how the PCA was performed, for the broad reader it seems necessary to further describe the rationale behind performing this analysis and more background as to how the comparison is done.

Response: We apologise for what was an inadvertent omission of a paragraph on the PCA in our method section. We have now added appropriate text. In addition, we provide further rationalisation for using PCA over conventional enrichment analysis of the pulldown data that we obtained. Please see the text in the supplementary material.

-Line 479: (Associated with figure 7E) The authors visualized only custom tracks, but do not provide count reads. Since there is a possibility that EB1 transcripts are still produced by the EB1-KO parasites it would be good to also provide a direct link for custom tracks in order for the reader to investigate the reads coverage thoroughly. Further, the authors did not provide an indicator for the adj P-value in Fig. 7F

Response: The read count range for Fig. 7E is 0 to 900 reads and this information has been added to both the figure and figure legend. Genome browser tracks can now be found at github.com/Sabel14/EB1_RNAseq_tracks (2 replicates for each sample except EB1KO-0 min, which had only one successful run and was not used for RNA-seq analysis). The adjusted P-value cutoff (0.05) for Fig 7F has been added to the figure, figure legend, and the results section.

Line 481: It would be good to emphasize how the GO enrichment analysis was performed in this section (Note Fig S10 is absent). Line 1329: Fig S10 cannot be found in the supplementary materials.

Response: Information on how the GO test was done has been added. We apologise for the omission of Fig S10 which has now been added to the revised version-

REVIEWERS' COMMENTS

Reviewer #1 (Remarks to the Author):

In the revised manuscript, the authors have mostly addressed the previous comments from this and the other reviewers. The revised manuscript is easier to follow and the text more clearly matches the data.

I have no additional major comments.

Reviewer #2 (Remarks to the Author):

The authors have addressed my queries.

Reviewer #3 (Remarks to the Author):

I believe that the authors have addressed my questions and concerns sufficiently and I have no further comments to add.